# Phylogeography and Population History of *Eleutharrhena macrocarpa* (Tiliacoreae, Menispermaceae) in Southeast Asia's Most Northerly Rainforests

**Shijie Song [1,2], Jianyong Shen [1], Shishun Zhou [1], Xianming Guo [3], Jinchao Zhao [4], Xinghui Shi [5], Zhiyong Yu [6], Qiangbang Gong [7], Shaohua You [8] and Sven Landrein [1,*]**

1 Xishuangbanna Tropical Botanical Garden, Chinese Academy of Sciences, Xishuangbanna 666303, China; songshijie@xtbg.ac.cn (S.S.); shenjianyong@xtbg.ac.cn (J.S.); zss@xtbg.org.cn (S.Z.)
2 College of life sciences, University of Chinese Academy of Sciences, Beijing 101408, China
3 Research Institute of Xishuangbanna National Nature Reserve, Xishuangbanna 666100, China; xianmingguo56@gmail.com
4 Management Bureau of Nangunhe National Nature Reserve, Lincang 677400, China; jinchaozhao91@gmail.com
5 Pingbian Branch of Management and Protection Bureau of Daweishan National Nature Reserve, Honghe Hani and Yi Autonomous Prefecture 661299, China; xinghuishi26@gmail.com
6 Management Bureau of Fenshuiling National Nature Reserve, Honghe Hani and Yi Autonomous Prefecture 661500, China; zhiyongyu41@gmail.com
7 Tongbiguan Provincial Nature Reserve, Dehong Dai and Jingpo Autonomous Prefecture 678400, China; qiangbanggong@gmail.com
8 Xishuangbanana National Nature Reserve, Xishuangbanna 666100, China; youshaohua266@gmail.com
* Correspondence: sven@xtbg.ac.cn

**Abstract:** The diversification of Tiliacoreae and the speciation of *Eleutharrhena* are closely linked to Southeast Asia's most northerly rainforests which originate from the Himalayan uplift. Migration routes across biogeographical zones within the Asian clade, including those of *Eleutharrhena*, *Pycnarrhena*, and *Macrococculus*, and their population structures are still unexplored. We combine endocarp morphology, phylogenetic analyses, divergence time estimation, ancestral area reconstruction, as well as SCoT method to reconstruct the past diversification of *Eleutharrhena macrocarpa* and to understand their current distribution, rarity, and evolutionary distinctiveness. The disjunct, monospecific, and geographically restricted genera *Eleutharrhena* and *Macrococculus* both have a dry aril, a unique feature in Menispermaceae endocarps that further confirms their close relationship. *Pycnarrhena* and *Eleutharrhena* appeared during the end of the Oligocene c. 23.10 million years ago (Mya) in Indochina. *Eleutharrhena* speciation may be linked to climate change during this time, when humid forests became restricted to the northern range due to the Himalayan uplift. Differentiation across the Thai–Burmese range could have contributed to the isolation of the Dehong populations during the Miocene c. 15.88 Mya, when exchange between India and continental Asia ceased. Dispersal to the Lanping–Simao block and further differentiation in southeastern and southern Yunnan occurred during the Miocene, c. 6.82 Mya. The specific habitat requirements that led to the biogeographic patterns observed in *E. macrocarpa* contributed to a low genetic diversity overall. Population 1 from Dehong, 16 from Pu'er, and 20 from Honghe on the East of the Hua line have a higher genetic diversity and differentiation; therefore, we suggest that their conservation be prioritized.

**Keywords:** aril; conservation genetics; *Eleutharrhena macrocarpa*; evolutionary distinctiveness; Himalayan uplift; Menispermaceae; southern Yunnan rainforests; Tiliacoreae

## 1. Introduction

*Eleutharrhena macrocarpa* (Diels) Forman is a woody liana belonging to the Tiliacoreae tribe in the Menispermaceae or moonseed family. It is primarily distributed across China (southern Yunnan), Laos, Myanmar, and India [1] (Figure 1). Like many species of the

moon-seed family, members of the Tiliacoreae are typically dioecious lianas inhabiting tropical rainforests and monsoon forests in South America, Africa, Asia, and Oceania [2]. Tiliacoreae Miers sensu Ortiz [2] includes 16 genera and 111 species, among which 6 genera are monospecific, including *Eleutharrhena* [2–4]. Although the sister relationships within the Tiliacoreae are weakly supported, the monophyly of the tribe is well established (24 species, 10 genera, and 3 plastid regions were used in analyses by Ortiz, Wang, Jacques and Chen [2], whereas 26 species, 11 genera, and 7 plastid plus 2 nuclear regions were used in the analyses by Lian, et al. [5]). The tribe is morphologically characterized by male flowers with more than four whorls of sepals, longitudinally grooved seed endocarps that are ribbed or rugose abaxially, hippocrepiform seeds without endosperm, and subcylindrical embryos [2]. Seeds play an important part in the classification of Menispermaceae and are highly diagnostic [6]. Fruits of Menispermaceae have woody endocarps and the seeds are often curved around an invagination of the lateral sides, termed the condyle [7]. The origin of the condyle is described by Ortiz [7] as resulting from the development of the ovary wall. Two main types of condyle were identified. Only the *Menispermum* condyle type occurs in Tiliacoreae, which have deeply curved seeds resulting from a bilaterally compressed, laminiform condyle.

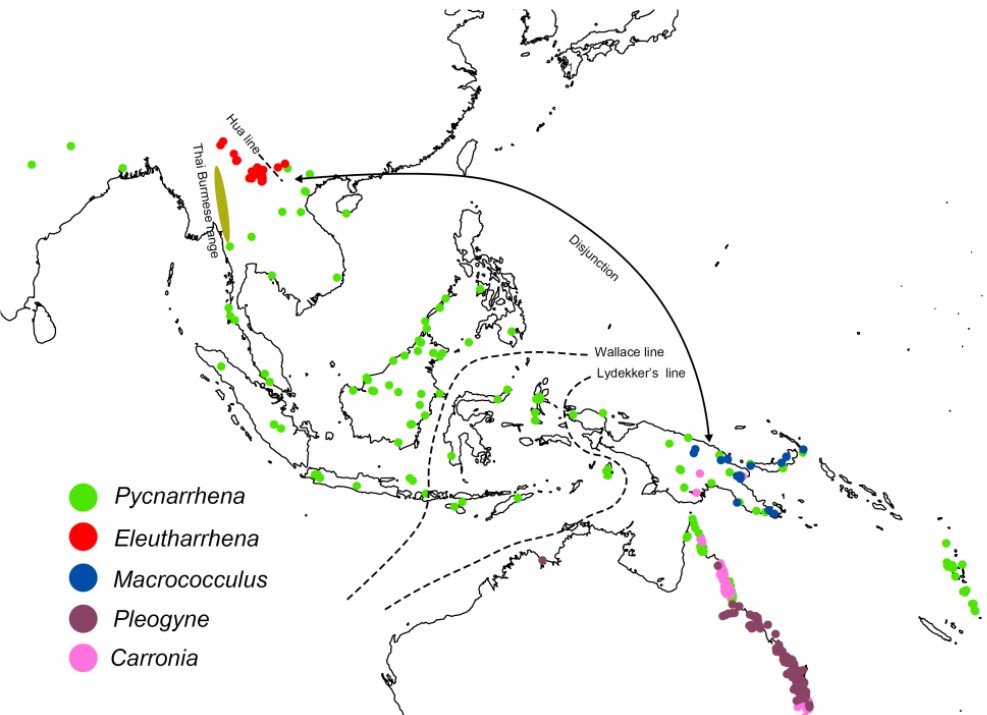

**Figure 1.** Distribution map of *Eleutharrhena* and sister genera *Macrococculus*, *Pycnarrhena*, *Pleogyne*, and *Carronia*. Distribution records downloaded from GBIF except for *Eleutharrhena macrocarpa* records.

*Eleutharrhena* is a sister to *Pycnarrhena* [5] and is morphologically similar to *Macrococculus* [8]. Their ranges overlap, but *E. macrocarpa* morphologically differs by its free stamens, stipitate drupelets, and grouped stomata on the abaxial leaf surface. It can become a large woody vine reaching the canopy and produces very large drupelets, although these are not as large as those of *Macrococculus*, which are said to be dispersed by flightless cassowary birds (*Casuarius)* [9]. *Eleutharrhena macrocarpa* was assessed to be a critically endangered species [10] and a plant species with extremely small populations (PSESP) in China [11]. Hou et al. [12] reported approximately 40 plants surviving in Yunnan, China, and Lang et al. [1] estimated that 60 individuals occurred in Yunnan and we sampled 48 individuals in our study (Figure 2).

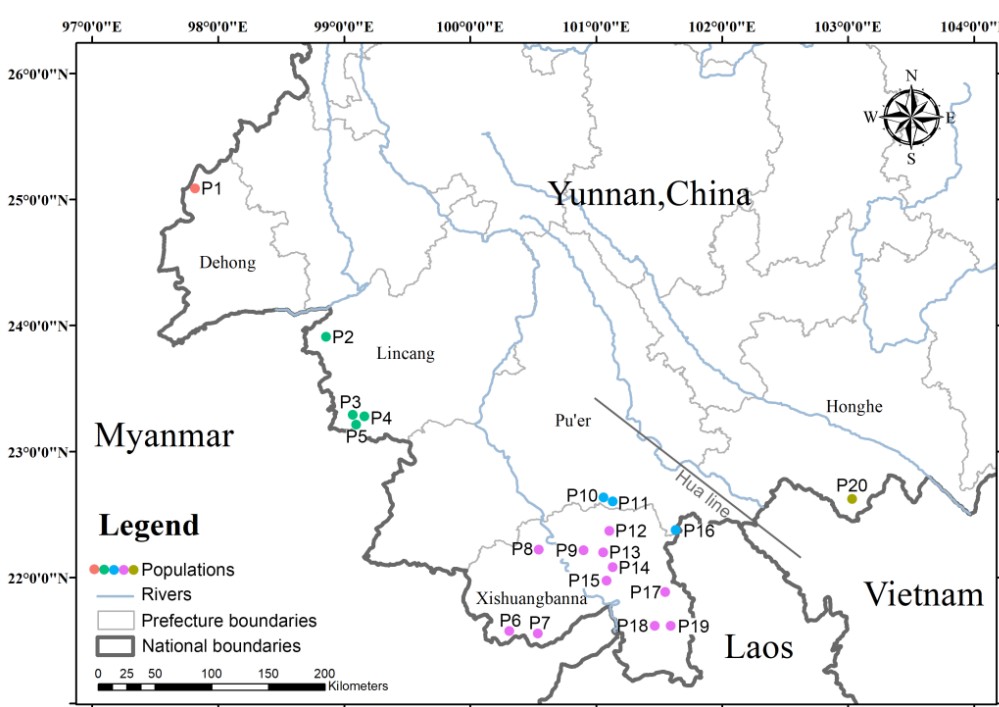

**Figure 2.** Distribution map of *Eleutharrhena macrocarpa* populations used for genetic diversity and structure analysis.

The origin of Menispermaceae has been evaluated to be c. 109 Mya in the Indo-Malayan region and it was suggested that the radiation of the family is linked to the simultaneous appearance of modern neotropical and paleotropical forests around the world following the mass extinction at the Cretaceous–Paleogene boundary [13]. The tropical rainforests in southern China may have appeared during the late Tertiary accompanied by the uplift of the Himalayan mountains and the development of a monsoon climate [14]. The structure of these forests is similar to that of SE Asian lowland rainforests, but they have a deciduous tree layer, fewer megaphanerophytes and epiphytes, and more abundant lianas [14].

Several distinct floristic regions and biogeographical areas can be recognized in tropical southern China: Taiwan, Hainan, part of Guangxi, southeastern Yunnan, and southern Yunnan [14]. The movement of the Indochina geoblock and the Lanping–Simao block together with the Himalayan uplift has profoundly influenced the flora of southern Yunnan [15]. The floras of southern and southeastern Yunnan have a high proportion of tropical Asian elements, but the southern part is more closely related to the Indo-Malaysian flora, suggesting that a boundary exists between these two parts [15]. *Eleutharrhena* is a typical Indo-Malaysian element and its distribution may closely correspond to the biogeographical events that led to the formation of southern Yunnan flora. Several lines have been proposed to explain floral boundaries in Yunnan; the Hua line [15] between southern and southeastern Yunnan floras; the Tanaka line [16] which was used to describe the difference between the Sino Himalayan forests to the west and the Sino Japanese forests to the east; and the Salween–Mekong divide, separating the east Himalayan from the Hengduan mountains during glacial periods [17]. However, the influence of these different lines and the period of occurrence may not all apply to the tropical flora of southern Yunnan or may only be valid for a part of it.

Although Menispermaceae are closely linked to the expansion of tropical rainforests and their speciation could be linked to interglacial periods as well as distinct biogeographic events, few studies have reported genetic structures at the species level. Examples include the distribution of *Sinomenium acutum* (Thumb.) Rehd. et Wils. in subtropical China using chloroplast marker haplotypes [18], the genetic structure of *Stephania yunnanensis* H.S.Lo

using DALP [19], and the link between *Chasmanthera dependens* Hochst. climatic refugia and genetic diversity in West Africa using AFLP and plastid markers [20].

In this study, we combine phylogenetics, time-measured phylogenies, ancestral area reconstruction, and population genetics based on the Start Codon Targeted (SCoT) polymorphism methods to answer questions about the phylogeography and the population history of *Eleutharrhena macrocarpa*. Specifically, we investigate how the genetic diversity is linked to the diversification of the southern Yunnan flora, the Himalayan mountain uplift, interglacial periods, and the occurrence of distinct population partitions. Furthermore, we focus on the seed morphology and its evolution within Tiliacoreae. Finally, we provide data on the populations' genetic structure that can be used to prioritize populations for conservation.

## 2. Materials and Methods

### 2.1. Morphological Observations and Analysis

One fruiting specimen of *Eleutharrhena macrocarpa* was selected and fruits were boiled for three hours and then left to soak overnight. Endocarps were then dissected and observed under a stereomicroscope (Nikon corporation). For comparison with other seeds having a strongly curved embryo, a conspicuous condyle and intrusive tissue such as the raphe, we also dissected the fresh fruits of *Haematocarpus validus* (Miers) Bakh.f. ex Forman (Pachygoneae). *Haematocarpus* is often misidentified as *E. macrocarpa* in the field but markedly differs in its seed morphology [5]. The specimens were collected in Menglun forest reserve in Xishuangbanna, Yunnan. For description and terminology, we followed Wefferling, Hoot and Neves [6]. Endocarp characteristics were mapped on the Bayesian phylogenetic tree (see 2.3) using Mesquite v. 3.70 (https://www.mesquiteproject.org/ accessed on 5 May 2022) to observe the changes in character states. A maximum likelihood (ML) analysis was performed to determine the ancestral character states. Trees were visualized using the ball and stick method, with pie charts at each node indicating the proportional likelihood of ancestral characteristics. A heuristic search with Maxtrees set to 100 and a majority rule consensus tree based on 17 morphological characters [3] (Table A1) was also performed for all the genera in Tiliacoreae using Mesquite v. 3.70.

### 2.2. Sampling, DNA Extraction, PCR, and Electrophoresis

Thirty-three genetic sequences and one outgroup [21] from Tiliacoreae were downloaded from the National Center for Biotechnology Information (NCBI) for the phylogenetic study. Seven plastid regions (*matK*, *ndhF*, *rbcL*, *atpB*, *trnL-F*, *trnH-psbA,* and *rps16*) and two nuclear regions (*26S rDNA* and *ITS*) were selected according to several recently published Menispermaceae phylogenetic studies [2,5,22] (see Table A2). One additional DNA sample collected in Xishuangbanna was extracted and its whole chloroplast was sequenced using NGS, aligned, assembled, and annotated by us (GenBank: MZ502223, detailed methods not shown here). The regions of interest were then extracted using Geneious v. 8.1.3 and added to our matrix. The sequences were aligned using the Muscle Algorithm [23] and the resulting sequences were edited with Geneious v. 8.1.3.

Silica-dried leaves of 48 individuals were collected in China for the SCoT analysis. Each population (from Dehong to Honghe) was more than 4 km apart. DNA was extracted using the Geneon Plant Genomic DNA Kit (Geneon, Inc. Changchun, China) and five geographically distant samples were selected to test 36 universal SCoT primer pairs [24]. PCR amplifications were conducted in a 50 µL volume, including 45 µL Tsingke master mix, 4 µL primers (10 µM), and 1 µL DNA extract. A C1000 Touch thermocycler (Bio-Rad Laboratories, Inc. China) was used to perform amplifications with the following program: 95 °C for 5 min; 35 cycles at 98 °C for 30 s, 52 °C for 30 s, 72 °C for 1 min, and 72 °C for 5 min, and finally held at 4 °C. An ABI-3730xl Sequencer (Applied Biosystems, Ltd. UK) was used to perform the capillary electrophoresis using 5′-labeled fluorescent primers. Electrophoresis results were transferred to a 0–1 matrix for further analysis.

### 2.3. Phylogenetic Analysis, Dating, and Ancestral Area Reconstruction

Phylogenetic trees were inferred with maximum likelihood (ML) and Bayesian inference (BI). ML analyses were performed using RAxML-HPC v. 8 on XSEDE [25] using the CIPRES Science Gateway v. 3.3 portal [26] based on GTR-CAT model with 1000 replications. The Bayesian analysis was implemented with MrBayes v. 3.2.3 [27] in PhyloSuite v. 1.2.2 platform [28] specifying the DNA substitution model selected by partitionFinder2 [29]. We performed four runs using 10 million generations with four chains, sampling trees every 1000 generations. The first 20% of trees were discarded and a 50% majority rule consensus tree was reconstructed from the remaining post-burn-in trees. The average standard deviation of the split frequency was used to verify that all runs reached stationarity and converged on the same distribution. To estimate the divergence time in Menispermaceae and Tiliacoreae, we ran a Bayesian relaxed molecular clock analysis with the plastid and nuclear combined dataset in BEAST v. 2.6.6 [30]. Our partitions were set in BEAUTi part of the BEAST package, and one fossil calibration point was inserted following Lian et al. [21] (*Triclisia inflata* 17.7 Mya can be assigned to extant *Triclisia* [31]). The parameters were the relaxed clock with a lognormal distribution and the Yule model. The maximum age of the Tiliacoreae root was set to 49.3 Mya [13] with a normal distribution and standard deviation of three. One run with 100 million generations and sampling every 5000 generations was conducted in BEAST. The first 10% of trees were discarded as burn-in. We used Tracer v. 1.7 [32] to assess the convergence and effective sample size. Tree Annotator v. 1.8.0 (part of the BEAST package) was used to summarize the set of post-burn-in trees and their parameters. FigTree v. 1.4.2 (http://tree.bio.ed.ac.uk/software/figtree/ accessed on 1 February 2021) was used to visualize the tree and divergence times. Tiliacoreae clade (2), including *Eleutharrhena* and its sister genera, was extracted from the maximum clade credibility tree inferred with BEAST, and input in BioGeoBEARS [33] implemented in RStudio v. 1.4.1717 [34]. Four major geographical ranges were designed: Indochina, southern Yunnan, Sundaland including Wallacea and Australasia including New Guinea. Six models were tested, namely DEC, DEC + J, DIVALIKE, DIVALIKE + J, BAYAREALIKE, and BAYAREALIKE + J [33], and the most fitting model was selected by calculating the best value for the Likelihood Ratio Test (LRT). The resulting probabilities of the ancestral states were drawn as pie charts at the node on the provided tree.

### 2.4. Population Genetic Analysis

POPGENE v. 3.2 [35] was used to analyze the observed number of alleles (Na), the effective number of alleles (Ne), Shannon's information index (I), the percentage of polymorphic loci (PPL), and Nei's genetic diversity index (H). Genetic distance was visualized and interpreted using a PCoA analysis in GenAIEx v. 6.1 [36]. STRUCTURE v. 2.3.4 [37] was used to infer population structure following the Falush, et al. [38] input method, with a burn-in period of 100,000 iterations, 1,000,000 Markov Chain Monte Carlo (MCMC) repetitions, 1–5 K ranges, and 10 independent runs. Evanno's method [39] was used to determine the best number of subpopulations in the Structure Harvester v. 0.6.94 [40]. CLUMPP v. 1.1.2 [41] was used to align clusters and Distruct v. 1.1 [42] was used to visualize the results. Populations including more than one individual were used to infer genetic differentiation using phiPT ($\Phi$pt), a measure that allows intra-individual variation to be suppressed and is therefore ideal for comparing codominant binary data [43,44] and gene flow (Nm = [(1/$\Phi$pt) −1]). An analysis of molecular variance (AMOVA) was performed among populations and within populations. Mantel's test was performed to determine the relationship between genetic and geographic distances in GenAIEx v. 6.51.

## 3. Results

### 3.1. Endocarps

The druplets of both *Eleutharrhena macrocarpa* and *Haematocarpus validus* are very similar in terms of size, number, and the stipitate base. The exocarps of *H. validus* (Figure 3) differ by their deep red and fleshy cells, as well as the position of the style scar, which

is basal and not subapical as in *E. macrocarpa* (Figure 3). Endocarps have a deep dorsal longitudinal groove as well as several lateral grooves in *H. validus*, whereas only a ventral ridge and faint transversal ridge can be seen on the lateral side of *E. macrocarpa* endocarps. Two large extruded funicle apertures can be observed at the base of the ventral part in *E. macrocarpa*. The endocarp of *H. validus* is clearly deeply curved with a pronounced bilaterally compressed condyle, which is absent in *E. macrocarpa*, and the raphe is clearly visible and intrusive inside the condyle in *H. validus*. Both genera have seeds without endosperm and fleshy cotyledons, although *E. macrocarpa* cotyledons are unequal. *E. macrocarpa*'s endocarps are more similar to *Pycnarrhena*'s but markedly differ by the presence of two basal and ventral funicle apertures and the presence of a well-developed bilobed aril surrounding the cotyledons (Figure 3). We could not obtain the fruit of *Macrococculus*, but Forman [8] observed that the seed is covered with a reticulate membrane as well as two basal and ventral funicle apertures and is thus very similar to *Eleutharrhena*.

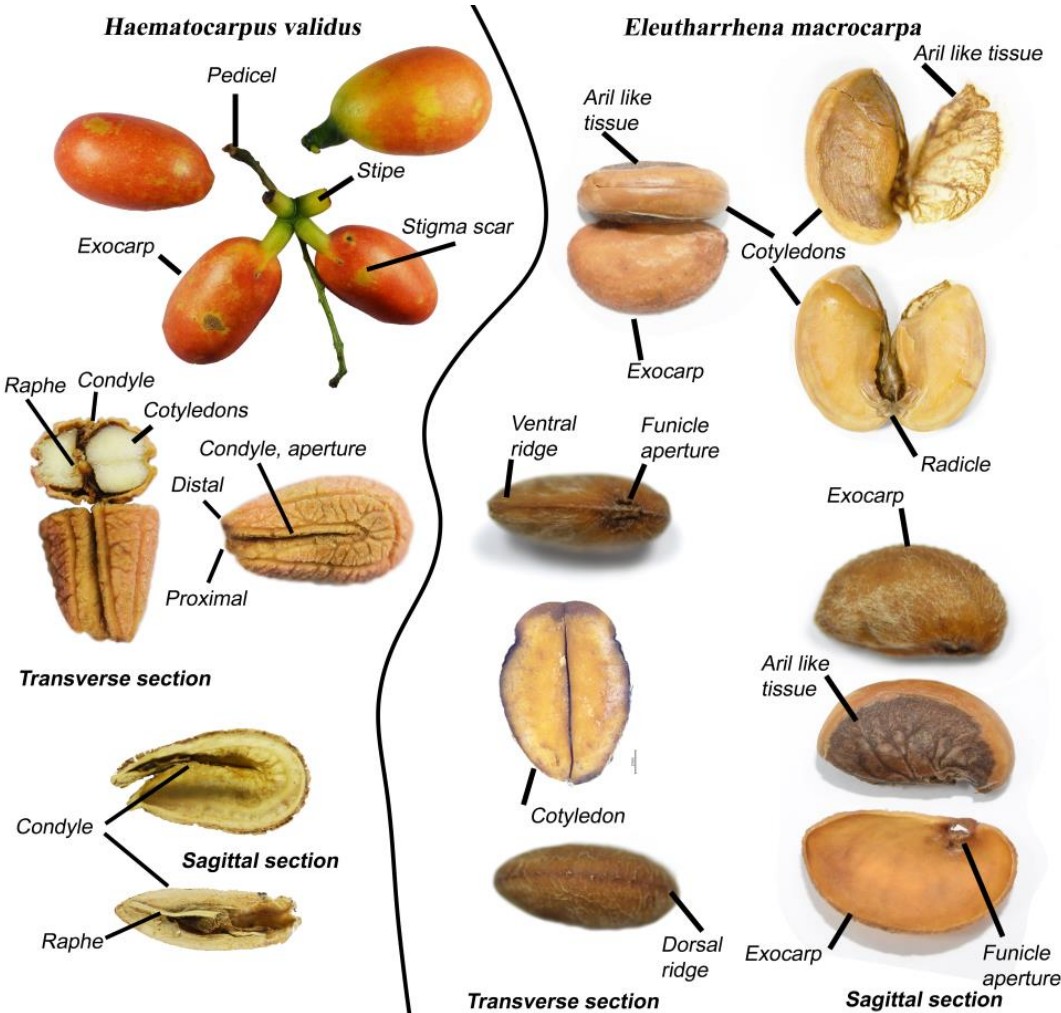

**Figure 3.** Fruits morphology of *Haematocarpus validus* and *Eleutharrhena macrocarpa* (sagittal and transverse sections).

The node at the base of the Tiliacoreae indicates a 99% maximum likelihood (ML) that the common ancestor had endocarps with deeply curved embryo and a conspicuous condyle. The node at the base of clade (2) had a 95% ML for a deeply curved embryo with intrusive condyle. Additionally, the node at the base of *Pycnarrhena* and *Eleutharrhena* had a 45% ML for weakly curved embryo and weak condyle and a 43% ML for a straight embryo with an intrusive aril (Figure 4). Based on 17 morphological characters (Table A1),

the heuristic search produced an incongruent tree with the molecular data (Figure A1). However, *Macrococculus* and *Eleutharrhena* had a 98% support for their sister relationship.

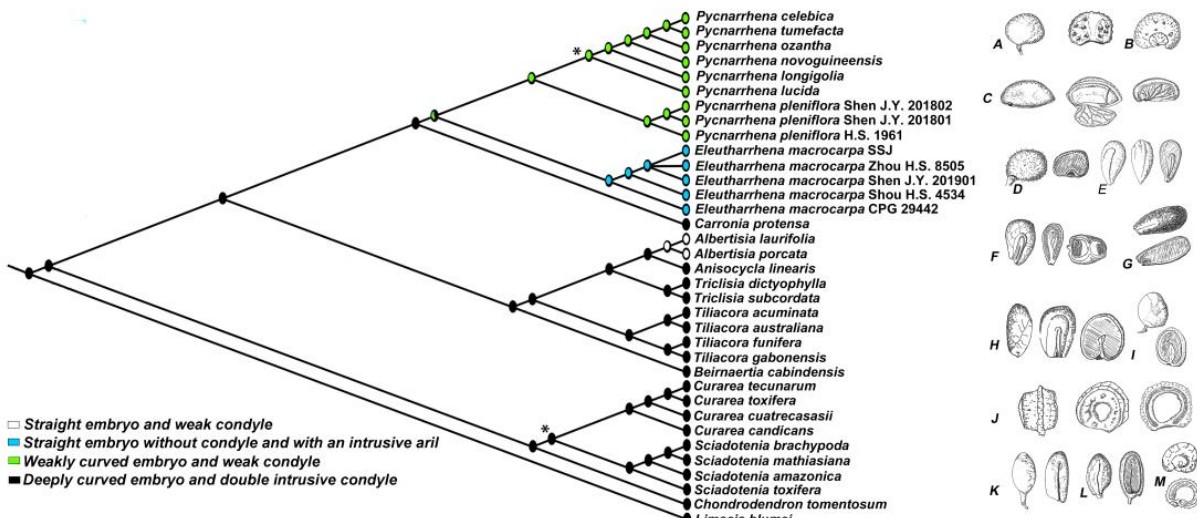

**Figure 4.** Endocarp characteristics mapped on the Bayesian phylogenetic tree of Tiliacoreae using mesquite. All nodes have more than 0.9 posterior probabilities (PP) support except when noted with *, meaning less than 0.8 PP. Endocarp illustrations: A. *Pycnarrhena pleniflora*; B. *Pycnarrhena ozantha*; C. *Eleutharrhena macrocarpa*; D. *Pleogyne cunninghamii*; E. *Carronia thyrsiflora*; F. *Tiliacora triandra*; G. *Albertisia scandens*; H. *Anisocycla jollyana*; I. *Beirnaertia cabindensis*; J. *Syrrheonema fasciculatum*; K. *Curarea* sp.; L. *Chondrodendron platyphyllum*; M. *Sciadotenia pubistaminea*. D and J were not sampled and added to the analysis because only a few DNA regions were available and their branches had no support.

### 3.2. Phylogenetic Analyses

The matrix was 10657 bp-long, including 469 parsimony-informative characters. The ML and Bayesian analyses of the plastid and nuclear-combined dataset were mostly congruent (see Figure A2). Three groups within Tiliacoreae can be identified: (1) the neotropical genera *Curarea*, *Sciadotenia*, and *Chondrodendron* (BS = 100%, PP = 1); (2) the Asian and Australasian genera *Carronia*, *Eleutharrhena*, and *Pycnarrhena* (BS = 95%, PP = 0.99); and (3) the remaining paleotropical and Australasian genera *Tiliacora*, *Albertisia*, *Anisocycla*, *Triclisia*, and *Beirnaertia* (BS = 80%, PP = 0.99).

### 3.3. Dating Analyses

The maximum clade credibility tree of the plastid and nuclear combined alignment using the Yule model is displayed in Figure 5. The neotropical clade (1) crown speciated during the Oligocene c. 31.19 Mya (95% highest posterior density interval (HPD): 23.13–39.73 Mya). The mostly African clade (3) diverged from clade (2) during the Eocene c. 40.06 Mya (95% HPD: 33.28–46.67 Mya). The differentiation between *Pycnarrhena* and *Eleutharrhena* occurred during the Oligocene c. 23.10 Mya (95% HPD: 17.45–29.57 Mya), whereas one individual of *Eleutharrhena macrocarpa* from Dehong diverged much earlier than the other individuals during the Miocene c. 15.88 Mya (95% HPD: 10.43–22.01 Mya). *Carronia* diverged from *Eleutharrhena* and *Pycnarrhena* during the Eocene c. 35.87 Mya (95% HPD: 27.93–43.57 Mya).

### 3.4. Ancestral Area Reconstruction of Eleutharrhena and Pycnarrhena

BAYAREALIKE + J model (Bayesian approach with a large number of areas under ML) had higher statistical support (see Table A3). The ancestral range of *Eleutharrhena* and *Pycnarrhena* was Indochina c. 23.10 Mya. The ancestral range of *Eleutharrhena* was Indochina c. 15.88 Mya and it dispersed to southern Yunnan c. 6.82 Mya (Figure 5). The

ancestral area of *Pycnarrhena* was Indochina with a dispersal to Australasia c. 15.51 Mya and finally Sundaland c. 2.16 Mya.

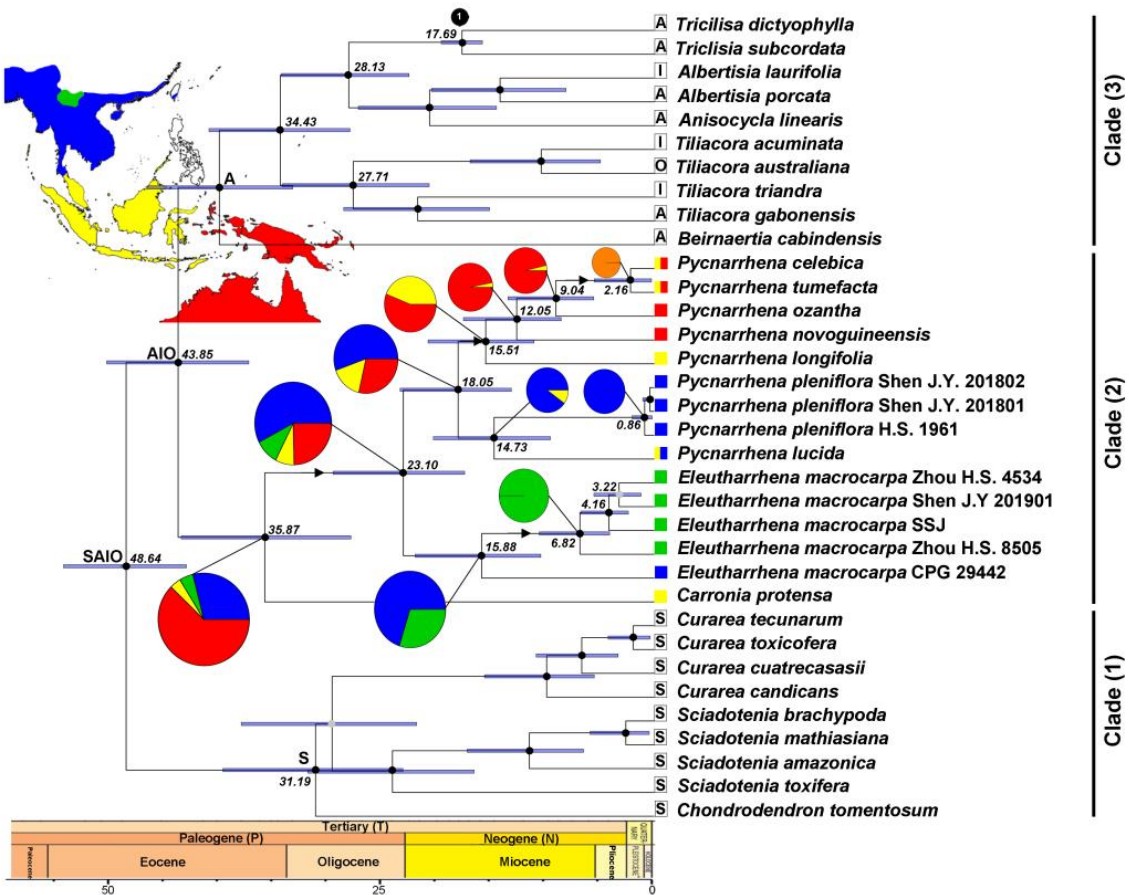

**Figure 5.** Chronogram of tribe Tiliacoreae resulting from the analysis of the plastid and nuclear combined dataset using the Yule model in BEAST 2. Nodes with black circles have posterior probabilities (PPs) between 0.9 and 1; grey circles have a PP between 0.6 and 0.8. Blue bars correspond to the 95% highest posterior density intervals (HPD) for the corresponding node. Fossils: 1. *Triclisia inflata* 17.7 Mya. Ancestral range reconstruction of tribe Tiliacoreae using the package BioGeoBEARS in R. The best model for clade (2) was BAYAREALIKE + J with 4 areas max, d = 0.003, e = 0, j = 0.074 and LnL = −21.97. Arrows indicate a dispersal event. Areas for *Eleutharrhena* and *Pycnarrhena*—red: Australasia; blue: Indochina; yellow: Sundaland; green: southern Yunnan; orange: Australasia + Sundaland. Areas for the remaining genera in Tiliacoreae: S: South America; A: Australasia; I: Indochina; O: Australasia.

*3.5. Population Genetic Diversity*

SCoT primers 7, 8, and 23 were the most polymorphic, producing unambiguous clear bands and were subsequently selected for amplification and analysis. Forty-eight individuals generated 536 bands in total. Genetic diversity indices for populations including more than one individual are presented in Table 1. Nei's genetic diversity index (H) ranged from 0.228 to 0.3120, and Shannon's information index (I) ranged from 0.335 to 0.4715. Populations 20, 1 and 16, located in Honghe, Dehong and Pu'er, respectively, had the highest genetic diversity (H = 0.3120, I = 0.4715; H = 0.3062, I = 0.4567 and H = 0.297, I = 0.4506, respectively). The mean genetic diversity value (H = 0.2705, I = 0.4106) is low, the gene flow ($N_m$ = 0.6983) is medium according to the standard used in [45] and the coefficient of gene differentiation $G_{ST}$ (among populations) = 0.4173. All these values show a high population differentiation [46]. The genetic diversity was H = 0.169 and I = 0.2873 for all 48 individuals.

**Table 1.** Genetic diversity of *Eleutharrhena macrocarpa* populations.

| Populations | Individuals | Na | Ne | H | I | PPL |
|:---:|:---:|:---:|:---:|:---:|:---:|:---:|
| P1 | 3 | 1.8217 | 1.5187 | 0.3062 | 0.4567 | 82.2% |
| P4 | 2 | 1.5539 | 1.3917 | 0.2294 | 0.335 | 55.4% |
| P5 | 2 | 1.6201 | 1.4385 | 0.2568 | 0.375 | 62.0% |
| P7 | 3 | 1.8267 | 1.4623 | 0.2866 | 0.4363 | 82.7% |
| P9 | 7 | 1.9349 | 1.4148 | 0.2612 | 0.4115 | 93.5% |
| P13 | 6 | 1.9487 | 1.4140 | 0.2618 | 0.4144 | 94.9% |
| P16 | 3 | 1.8468 | 1.483 | 0.297 | 0.4506 | 84.7% |
| P18 | 7 | 1.8968 | 1.3546 | 0.228 | 0.3663 | 89.7% |
| P19 | 2 | 1.6425 | 1.4543 | 0.2661 | 0.3885 | 64.3% |
| P20 | 3 | 1.8779 | 1.512 | 0.312 | 0.4715 | 87.8% |

Na = observed number of alleles; Ne = effective number of alleles; H = Nei's gene diversity; I = Shannon's information index; PPL = the percentage of polymorphic loci.

### 3.6. Cluster Analysis and Population Structure

The principal coordinate analysis (PCoA) including all samples shows that three subpopulations can be observed (Figure 6a). Subpopulation 1 has two individuals from Dehong. Individuals from Lincang are all in subpopulation 2. Subpopulations 2 and 3 are mixed between Xishuangbanna, Dehong, Honghe, Lincang, and Pu'er individuals. PCoA results indicate that subpopulations 2 and 3 are distinct, although some individuals from Xishuangbanna belong to subpopulation 2 or 3. Most individuals from southeastern Yunnan, eastern Xishuangbanna and Pu'er are grouped in subpopulation 3 and most individuals from western southern Yunnan and western Xishuangbanna are in subpopulation 2. No natural geographical barriers exist between subpopulations 2 and 3 except for the Lancang river. Structure analyses and Evanno's methods also indicated the best K = 3, and Honghe individuals have a similar admixture to that of the Xishuangbanna, Pu'er, and Lincang populations, whereas two individuals from subpopulation 1 in Dehong have more private alleles, as also reflected in the plastid phylogenetic analysis (Figure 6b).

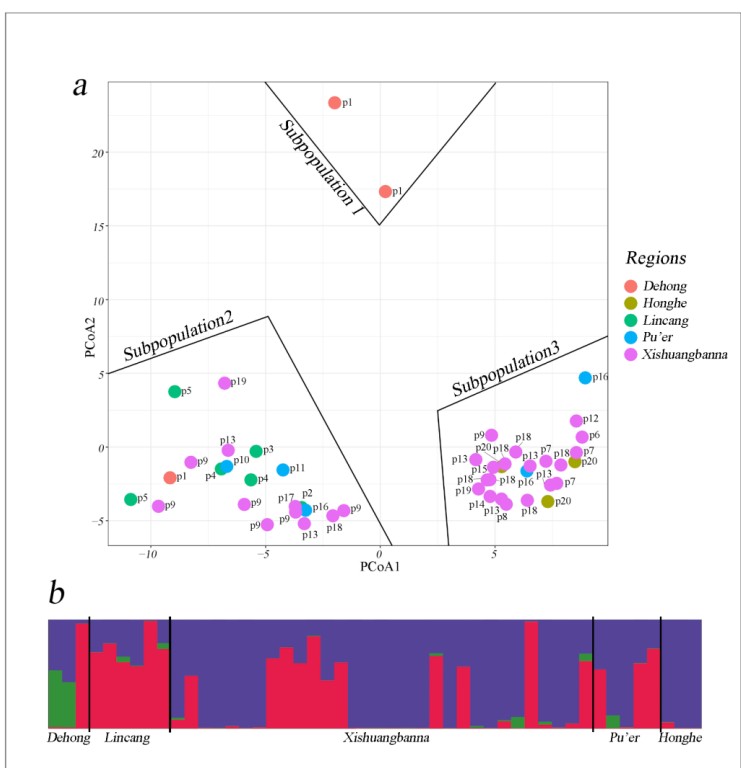

**Figure 6.** (**a**) Principal coordinates analysis of *Eleutharrhena macrocarpa* based on the genetic similarity matrix of SCoT markers; and (**b**) STRUCTURE analysis revealed that the best K is 3.

### 3.7. Analysis of Molecular Variance and Mantel's Test

The AMOVA results show that most of the genetic variation occurred within populations (83%) whereas only 17% of the variance occurred among populations (see Table A4). The Mantel test shows no correlation between the geographical and genetic distance ($R^2 = 0.2299$), (see Figure A3).

## 4. Discussion

### 4.1. Endocarp Morphology

Tiliacoreae is characterized by male flowers with more than four whorls of sepals, longitudinally grooved seed endocarp that are ribbed or abaxially rugose, hippocrepiform seeds without endosperm and subcylindrical embryos [2]. A lack of morphological data, especially for the flowers of both sexes and the fruit has resulted in the patchy morphological data analyses of Menispermaceae tribes [6]. Using the morphological characteristics listed in the synopsis of the family [3], very few morphological characters could be used to produce a tree congruent with the phylogenetic analyses based on seven plastid and two nuclear regions (Figure A1). However, hippocrepiform seeds without endosperm and a conspicuous condyle seem to be a shared primitive characteristic of the tribe, except within *Eleutharrhena*, *Pycnarrhena,* and *Albertisia* (Figure 4). In *Eleutharrhena* (as well as *Macrocculus*, not sampled in our phylogenetic analysis), the condyle has completely disappeared, and instead, an aril-like tissue is present (Figure 3). We hypothesize that the bilobed aril is derived from the aborted ovule and could represent the dry cotyledons that now enclose the remaining fertile ovule [47]. This character may also be linked to the intrusion of organs within the condyle, as shown by the raphe in *Haematocarpus validus* (Figure 3) and does not seem to have an ecological function. Further apomorphies include the perforate condyle in *Syrrheonema*, the straight endocarp and reduced condyle in *Albertisia* as well as the ruminate endosperm in *Tiliacora*. The later ruminate endosperm is said to be of secondary origin, as some *Tiliacora* species have no endosperm.

### 4.2. Tiliacoreae and Eleutharrhena Biogeography

Our phylogenetic results based on seven plastid and two nuclear regions are similar to previously obtained phylogenies [2,5] and confirm three clades within the tribe. The distribution of the tribe Tiliacoreae includes South America, Africa, Asia, and Australasia and resembles groups that have experienced western Gondwana vicariance, c. 105 Mya [48].

*Pycnarrhena* and *Eleutharrhena* appeared during the Oligocene c. 23.10 Mya in Indochina. During the late Eocene and Oligocene, the climate for much of Southeast Asia was seasonal and humid forests became isolated [49]. We can infer that this climatic change may have favored the speciation of *Eleutharrhena* in its northern range due to the Himalayan uplift and the apparition of a monsoon climate. *Pycnarrhena* dispersal to Australasia and Sundaland followed a distinctive pattern that can also be observed in *Tiliacora*—firstly that of a dispersal from Indochina to Australasia during the Miocene c. 15.51 Mya and then back to Sundaland c. 2.16 Mya. The main change at the beginning of the Miocene is the change in Sundaland from a seasonally wet to humid climate. This corresponds to the closure of the Indonesian throughflow, and the Australian plate collision with Southeast Asia [49]. Dispersal from Asia to Australia may have occurred when the sea level is known to have been lower, allowing for some migrations to happen across the Wallace line [50,51]. The Wallace line is a well-known biogeographical barrier between New Guinea/Australia and Sundaland [52], a deep trench that separated both floras and which could have separated earlier lineages. The disjunction of *Eleutharrhena* + *Macrococculus* (based on morphological data only) across the Wallace line may be explained by the narrow habitat requirements of these genera and is linked to the distribution of humid rainforests.

*Eleutharrhena macrocarpa* samples from Xishuangbanna diverged very early from the individuals from Dehong during the Miocene c. 15.88 Mya (Figure 5). Differentiation across the Thai–Burmese border range (Tenasserim Hills, Dawna Range, and Karen Hills) was proposed to explain the diversity in several bird and mammal species [53]. Molecular

data [54] suggest that exchanges between the Indian subcontinent and mainland Asia peaked from 44 Mya in the Eocene to the mid Miocene, and then decreased after 14 Mya due to the drier conditions developing in northern India. It is therefore possible that the Dehong population may represent a separate refugium across the Thai–Burmese range that was previously connected to northern India and later became separated.

### 4.3. Eleutharrhena Macrocarpa Population Structure

A relatively high number of bands was obtained from the SCoT analysis, although between 53712 and 78921 unigenes were detected in *Menispermum canadense* and *M. dauricum* [55]. Our total bands therefore represent between 0.7% and 1% of the total number of unigenes, which may be due to the use of fluorescent primers and capillary electrophoresis.

Menispermaceae species are known to be outbreeding species which should result in reduced population differentiation. *Sinomenium acutum* populations were shown to have a high gene flow and low population differentiation, although refugia populations were identified by their higher genetic diversity ($H_T$ = 0.828, $H_S$ = 0.710, $N_{ST}$ < $G_{ST}$, AMOVA within populations 83.62%, among populations 16.38% $N_m$ = 2.552) [18]. *Tinospora cordifolia* had relatively low genetic diversity but this study was only based on five individuals (H = 0.2114) [56]. *Menispermum canadense* and *M. dauricum* were found to have a relatively high inbreeding coefficient and low genetic diversity (inbreeding coefficient of 0.198, $H_o$ = 0.377, and $H_e$ = 0.342 and no significant deviation from the Hardy–Weinberg equilibrium) [55]. The genetic diversity of *Chasmanthera dependens* showed opposite results from the cpDNA haplotypes and AFLP results (high population differentiation and weak geographical correlation for cpDNA, with $F_{ST}$ = 0.797 and low population differentiation and strong geographical correlation for AFLP, with $F_{ST}$ = 0.064) [20]. *Stephania yunnanensis* showed high population differentiation and relatively low gene flow ($N_a$ = 1.6192; $N_e$ = 1.4001; H = 0.2298; I = 0.43401) [19]. The view that most Menispermaceae species are outbreeding species with weak population differentiation contrasts with these previous and recent studies and could also show that the differentiation may be due to refugia, the founder effect, geographical isolation, and selection pressure [57].

*Eleutharrhena macrocarpa* shows low genetic diversity but high population differentiation and medium gene flow. The low level of genetic diversity and medium gene flow among populations but high population differentiation might be due to mixed forces involving (1) small population sizes and specific habitat requirements along with (2) sympatric barriers such as mountain ranges or rivers and (3) pollinator specificity [58]. Fragmentation and small population sizes could explain the relatively low differentiation of the populations from Honghe at the eastern limit of the range across the Hua line. The Hua line [15] was used to describe the different flora composition between southern and southeastern Yunnan. It has been suggested that this could correspond to the movement of the Indochina geoblock as well as the Lanping–Simao block together with the Himalayan uplift [15]. However, this line does not seem to have strongly affected the dispersal of *E. macrocarpa* to southeastern China.

Finally, *Eleutharrhena* has highly specific habitat requirements [59] which may be reflected by the higher population differentiation despite its low genetic diversity.

### 4.4. Evolutionary Distinctiveness

Our results suggest low genetic diversity and high population differentiation in *E. macrocarpa* and indicate that populations p1, p16, and p20 should be prioritized for conservation because of their higher genetic diversity and differentiation. The conservation value can be assessed using several methods and priorities such as threats, economic benefits, environment quality, and phylogenetic distinctiveness [60]. Our results show that *Eleutharrhena* has a high phylogenetic distinctiveness value because it is morphologically related to *Macrococculus* both having an unusual aril-like structure surrounding the seed. It has a high biogeographic value, as shown by the disjunction between these two genera as well as the isolation of the Dehong population. This is the first time that we have

documented differentiation across the Thai–Burmese border range for any plant species in southern Yunnan. *Eleutharrhena macrocarpa* is a relict plant of the mighty rainforests that has colonized and persisted in the northern range of the Southeast Asian biodiversity hotspot.

## 5. Conclusions

*Eleutharrhena* and *Macrococculus* are morphologically similar and *Eleutharrhena* is a sister to *Pycnarrhena*. *Eleutharrhena* may have evolved during the Himalayan uplift during the Miocene c. 15.88 Mya and its speciation may be linked to changes in climate during this time, when humid forests became restricted to the northern range. Dispersal to the Lanping–Simao block and further differentiation between southeastern and southern Yunnan occurred during the Miocene c. 6.82 Mya. The conservation of species with rich evolutionary histories and that are linked to major biogeographic events such as the Himalayan uplift should be prioritized.

Further studies involving the sampling of *Macrococculus,* the remaining genera within tribe Tiliacoreae, and the use of codominant genetic markers would improve our understanding of such complex biogeographical patterns.

**Author Contributions:** Conceptualization, S.S. and S.L.; methodology, S.S. and S.L.; software, S.S. and S.L.; validation, S.S. and S.L.; formal analysis, S.S. and S.L.; investigation, S.S., J.S., S.Z., X.G., J.Z., X.S., Z.Y., Q.G. and S.Y.; data curation, S.S. and S.L.; writing—original draft preparation, S.S.; writing—review and editing, S.L.; visualization, S.S. and S.L.; supervision, S.L.; project administration, S.L.; funding acquisition, S.L. All authors have read and agreed to the published version of the manuscript.

**Funding:** This research was funded by the Anthony Hitchcock Species Recovery Fund, grant number GBGF 2020/44717 and Center for Gardening and Horticulture, Xishuangbanna Tropical Botanical Garden, Chinese Academy of Sciences, grant number PPKP014B06.

**Institutional Review Board Statement:** Not applicable for studies not involving humans or animals.

**Informed Consent Statement:** Not applicable.

**Data Availability Statement:** Not applicable.

**Acknowledgments:** The authors thank Jo Osborne for manuscript review and editing. Thanks to Bulong Nature Reserve, Yiwu Nature Reserve and Nanpenghe Nature Reserve as well as the Forest Resources Management Station of Jinuo Mountain for providing help in conducting the surveys.

**Conflicts of Interest:** The authors declare no conflict of interest.

## Appendix A

**Table A1.** Morphological characters and character states used in this study.

| No. | Morphological Characters | State 1 | State 2 | State 3 | State 4 | State 5 |
|---|---|---|---|---|---|---|
| 1 | Number of male sepals | 6 | 9 | 12 | >15 | - |
| 2 | Number of male petals | 0 | 3 | 6 | - | - |
| 3 | Number of stamens | 3 | 6 | 9 | >12 | - |
| 4 | Stamen fusion | Free | Basally fused | Fused | - | - |
| 5 | Anthers dehiscence | Transverse/oblique | Longitudinal | - | - | - |
| 6 | Connective | Absent | Present | - | - | - |
| 7 | Number of staminodes | 0 | 3 | 6 | - | - |
| 8 | Number of carpels | 3 | 6 | 9 | >12 | - |
| 9 | Condyle/seed | Absent/embryo straight or slightly curved | Absent/aril present, embryo straight | Weak/embryo slightly curved | double lameliform/embryo strongly curved | Perforate/embryo strongly curved |
| 10 | Endosperm | Absent | Present | - | - | - |
| 11 | Cauliflory | Yes | No | - | - | - |

**Table A1.** *Cont.*

| No. | Morphological Characters | State 1 | State 2 | State 3 | State 4 | State 5 |
|-----|--------------------------|---------|---------|---------|---------|---------|
| 12 | Pseudocorolla | Yes | No | - | - | - |
| 13 | Clustered stomata | Yes | No | - | - | - |
| 14 | Petals in female flower | Yes | No | - | - | - |
| 15 | Inflorescence | Cymose/fasciculate | Capituliform | Flowers single | - | - |
| 16 | Glands on petals | Yes | No | - | - | - |
| 17 | Coroniform sepals | Yes | No | - | - | - |

**Table A2.** DNA sequences used for maximum likelihood and Bayesian phylogenetic analyses (new sequences are labeled with an asterisk *.

| Taxon, voucher, locality, *rbcL*, *atpB*, *matK*, *ndhF*, *trnL-F*, *trnH-psbA*, *rps16*, *26S rDNA*, and *ITS*. |
|---|
| ***Albertisia laurifolia***, Hong Y.P. 99371 (PE), Hainan, FJ626590, HQ260813, EF143849, JN051700, MW633412, MW633442, MW621979, EF143841; ***Albertisia porcata***, McPherson 16678 (MO), Gabon, HQ260758, HQ260814, KX384047, EF624261, KX384110, MW633413, MW633443, MW621980, -; ***Anisocycla linearis***, Hong-Wa et al. 466 (MO), Madagascar, HQ260759, HQ260816, JN051805, EF624263, JN051739, MW633414, MW633444, MW621981, MW621214; ***Beirnaertia cabindensis***, Walters and Niangadouma 1267 (MO), Gabon, HQ260766, HQ260822, JN051811, EF624270, JN051745, MW633415, MW633445, MW621982, MW621215; ***Carronia protensa***, van der Werff and Gray 17049 (MO), Australia, HQ260769, HQ260825, JN051815, EF624274, JN051749, MW633416, MW633446, MW621983, MW621216; ***Chondrodendron tomentosum***, Ortiz et Vásquez 217 (AMAZ, MO), Peru, HQ260771, HQ260828, JN051818, EF624278, JN051752, MW633417, MW633447, KM364844, -; ***Curarea candicans***, Torke 310 (MO), Guyana, HQ260776, HQ260832, JN051824, EF624288, JN051758, MW633418, MW633448, -, MW621217; ***Curarea cuatrecasasii***, Ortiz and Aguilar 324 (INB, MO), Costa Rica, MW633353, MW633367, KX384061, EF624289, KX384124, MW633419, MW633449, -, MW621218; ***Curarea tecunarum***, Ortiz and Vásquez 214 (AMAZ, MO), Peru, MW633354, MW633368, KX384062, EF624290, KX384125, MW633420, MW633450, -, MW621219; **Curarea toxicofera**, Ortiz 184 (AMAZ, MO), Peru, FJ026480, FJ026420, KX384063, EF624291, KX384126, -, -, -, -; ***Eleutharrhena macrocarpa* CPG29442**, CPG29442 (PE), Yunnan, MW633358, MW633371, MW633383, MW633394, MW633404, MW633424, MW633454, MW621986, MW621223; ***Eleutharrhena macrocarpa* Shen J.Y. 201901**, Shen J.Y. 201901 (HITBC), Yunnan, MW633357, MW633370, MW633382, MW633393, MW633403, MW633423, MW633453, MW621985, MW621222; ***Eleutharrhena macrocarpa* SSJ**, SSJ (HITBC), Yunnan, MZ502223 *, MZ502223 *, MZ502223 *, MZ502223 *, MZ502223 *, MZ502223 *, MZ502223 *, -, -; ***Eleutharrhena macrocarpa* Zhou H.S. 4534**, Zhou H.S. 4534 (HITBC, PE), Yunnan, MW633355, MW633369, MW633380, MW633391, MW633401, MW633421, MW633451, -, MW621220; ***Eleutharrhena macrocarpa* Zhou H.S. 8505**, Zhou H.S. 8505 (HITBC, PE), Yunnan, MW633356, -, MW633381, MW633392, MW633402, MW633422, MW633452, MW621984, MW621221; ***Limacia blumei***, F. Jacques 07 (P), cult. West Java, JN051683, JN051879, JN051836, EF624309, JN051770, -, -, -, -; ***Pycnarrhena celebica***, F. Jacques 014 (P), cult. Bogor, FJ026503, FJ026443, JN051846, -, -, -, -, -, -; ***Pycnarrhena longiflora***, F. Jacques 015 (P), cult. Bogor, EU526993, EU526965, JN051845, JN051716, JN051780, MW633425, MW633455, MW621987, MW621224; ***Pycnarrhena lucida***, A.F.G. Kerr 17887 (P), Thailand, MW633359, MW633372, MW633384, MW633395, MW633405, -, MW633456, MW621988, MW621225; ***Pycnarrhena novoguineensis***, Gray 8794 (MO), Australia, HQ260795, HQ260851, JN051847, EF624326, JN051782, MW633426, MW633457, MW621989, MW621226; ***Pycnarrhena ozantha***, T. Haevermans 418 (P), New Caledonia, MW633360, MW633373, MW633385, MW633396, MW633406, -, MW633458, MW621990, MW621227; ***Pycnarrhena pleniflora* H.S. 1961**, Zhou H.S. 1961 (PE), cult. Xishuangbanna, MW633361, MW633374, MW633386, MW633397, MW633407, MW633427, MW633459, MW621991, MW621228; ***Pycnarrhena pleniflora* Shen J.Y. 201801**, Shen J.Y. 201801(HITBC), Yunnan, MW633362, MW633375, MW633387, MW633398, MW633408, MW633428, MW633460, MW621992, MW621229; |

**Table A2.** *Cont.*

*Pycnarrhena pleniflora* **Shen J.Y. 201802**, Shen J.Y. 201802 (HITBC), Yunnan, MW633363, MW633376, MW633388, MW633399, MW633409, MW633429, MW633461, MW621993, MW621230; *Pycnarrhena tumefacta*, CPG33074 (PE), Bogor, Indonesia, MW633365, MW633378, MW633390, MW633400, MW633411, MW633432, MW633464, MW621995, MW621232; *Sciadotenia amazonica*, Ortiz and Zárate 264 (AMAZ, MO), Peru, HQ260799, HQ260855, JN051851, EF624330, JN051786, MW633433, MW633465, MW621996, MW621233; *Sciadotenia brachypoda*, Ortiz and Farroñay 222 (AMAZ, MO), Peru, -, -, -, EF624331, -, -, -, -, -; *Sciadotenia mathiasiana*, Ortiz and al. 259 (AMAZ, MO), Peru, MW633366, MW633379, KX384078, EF624332, KX384142, MW633434, MW633466, -, MW621234; *Sciadotenia toxifera*, Ortiz and al. 231 (AMAZ, MO), Peru, HQ260800, HQ260856, KX384079, EF624333, KX384143, MW633435, MW633467, -, MW621235; *Tiliacora acuminata*, F. Jacques 11 (P), cult. West Java, JN051696, JN051895, JN051861, JN051730, JN051796, MW633436, MW633468, MW621997, MW621236; *Tiliacora australiana*, Gray 9132 (QRS), Australia, JN051697, JN051896, JN051862, JN051731, JN051797, MW633437, MW633469, MW621998, MW621237; *Tiliacora funifera*, D. Kenfack 2100 (MO), Ghana, FJ026512, FJ026452, JN051863, EF624340, JN051798, MW633438, MW633470, KM364880, MK288774; *Tiliacora gabonensis*, Walters and Niangadouma 1159 (MO), Gabon, JN051698, -, JN051864, EF624341, JN051799, -, MW633471, MW621999, MW621238; *Triclisia dictyophylla*, Kenfack and Zapfack 2038 (MO), Cameroon, HQ260810, HQ260866, JN051866, EF624344, JN051801, MW633439, MW633472, MW622000, MW621239; *Triclisia subcordata*, Kenfack2101 (MO), Ghana, HQ260811, HQ260867, JN051867, EF624345, JN051802, MW633440, MW633473, MW622001, MW621240.

**Table A3.** Results from BioGeoBEARS models testing.

| Model | Number of Parameters | LnL | d | e | j | AIC | AICc | AIC Weight |
|---|---|---|---|---|---|---|---|---|
| DEC | 2 | −25.14 | 0.007 | 0 | 0 | 54.29 | 55.29 | 0.074 |
| DEC + J | 3 | −23.91 | 0.005 | 0 | 0.042 | 53.83 | 56.01 | 0.052 |
| DIVALIKE | 2 | −24.29 | 0.010 | 0 | 0 | 52.59 | 53.59 | 0.174 |
| DIVALIKE + J | 3 | −24.18 | 0.008 | 0 | 0.02 | 54.36 | 56.54 | 0.040 |
| BAYAREALIKE | 2 | −23.76 | 0.004 | 0.04 | 0 | 51.53 | 52.53 | 0.296 |
| BAYAREALIKE + J | 3 | −21.97 | 0.003 | 0 | 0.074 | 49.95 | 52.13 | 0.362 |

AIC and AICc comparisons of different models of biogeographical range evolution and estimates. AIC, Akaike Information Criterion; d, dispersal; e, extinction; j, weight of jump dispersal/founder speciation; LnL, log-likelihood of the model.

**Table A4.** AMOVA results among populations and within populations.

| Source | df | SS | MS | Est. Var. | % |
|---|---|---|---|---|---|
| Among populations | 9 | 852.278 | 94.698 | 10.873 | 17% |
| Within populations | 28 | 1528.143 | 54.577 | 54.577 | 83% |
| Total | 37 | 2380.421 | | 65.449 | 100% |

Df = degree of freedom; SS = sum of squares; MS = mean squares; Est. Var. = estimate of variance; % = percentage of total variation.

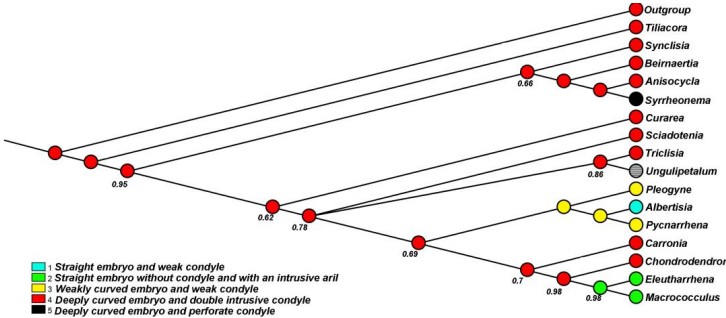

**Figure A1.** Heuristic search and majority rule consensus of Tiliacoreae based on 17 morphological characters using Mesquite. Endocarp ancestral character states mapped and visualized using the ball and stick method. Consensus tree supports 1 except when stated.

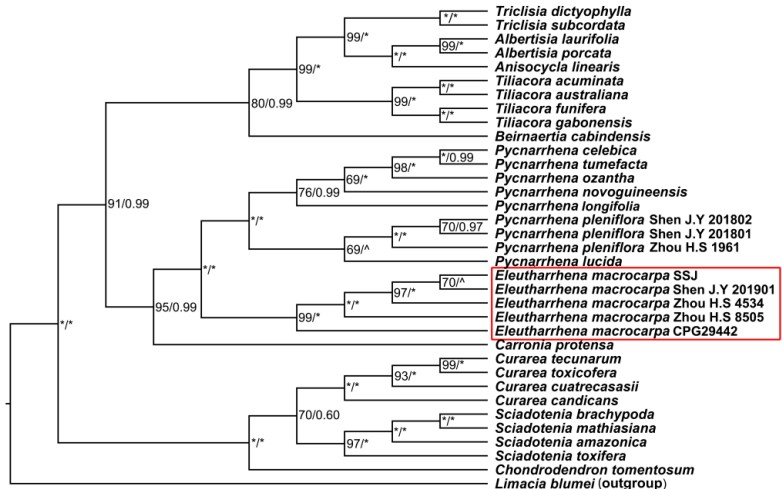

**Figure A2.** Maximum likelihood and Bayesian phylogenetic tree of the tribe Tiliacoreae based on the combination of *rbcL, atpB, matK, ndhF, trnL-F, trnH-psbA, rps16, 26S rDNA,* and *ITS* regions. * indicates BS = 100% or PP = 1.0; ^ indicates incongruency between ML and Bayesian phylogenetic trees.

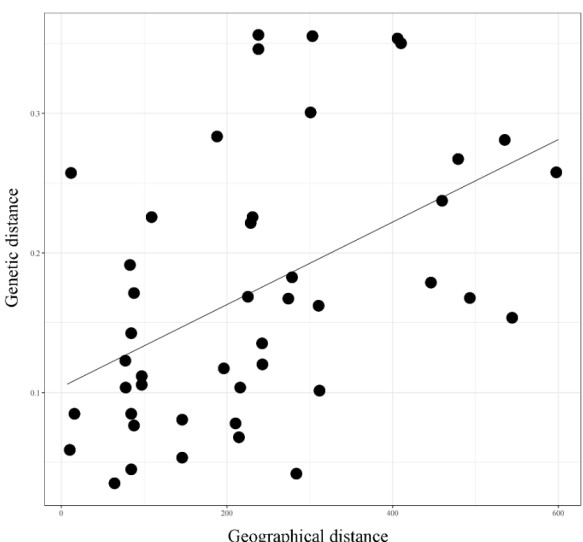

**Figure A3.** Mantel's test between genetic distance and geographical distance correlation. Equation: $Y = 0.0003\ x + 0.1029$, $R^2 = 0.2299$, $p = 0.01$.

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
