# Peer review of "Phylogeography and Population History of Eleutharrhena macrocarpa (Tiliacoreae, Menispermaceae) in Southeast Asia’s Most Northerly Rainforests"

_diversity, doi:10.3390/d14060437_

Round 1

Reviewer 1 Report

General comments

The manuscript presents an interesting biogeographic study, with phylogenetic and conservation implications.

Analyses are based on already published sequences, but one, for which the authors provide GenBank accession code (If I have correctly read; please, confirm).

I have no particular concern about the methods, results and interpretations.

However, the language must be carefully revised. I’m surprised that the text is full of errors/typos, given the presence of a native speaker among the co-authors. Many sentences are obscure.

Specific comments

86-94: A very messy sentence. Rephrase

95: possibly the authors mean: may not all apply to the tropical flora of southern Yunnan or may be valid only for a part of it.

80-81: add references to support this statement

86 between them is not clear, rephrase

123: You did not download species, but genetic sequences. Please, correct

Figure 3: explain colours in pie charts

260-61: this sentence is confused, please revise

304-305: this sentence is confused, please revise

307-308: this sentence is confused, please revise

309: does “where” refer to both genera, or only to Eleutharrhena. Please, revise to avoid ambiguity

Figure 5: explain the meaning of perf and no. Explain the meaning of the numbers and the scale bare.

358-9: confused; use (1) … (2) …(3)…. to clarify

365-368: revise punctuation or divide this long sentence into smaller ones

399-404:  confused, please revise (for example, it is not clear whether “which result” refers to specific habitat requirement; in this case result -> results). A full revision of this sentence is needed

406: provide references for the Hua line, the Tanaka line and the Salween Mekong divide

415-19: messy sentence; revise, please

425-430: messy sentence; revise, please

430-432: something is lacking. That have colonized what? When?

Typos/grammar

The manuscript is full of typos.

The tribe name Tiliacoreae is misspelled as Tiliacorea in various parts, even in the title.

Just a few examples of typos/grammar issues

42: Tiliacorea -> Tiliacoreae

44: delete comma after  [1]

45: tiliacoreae -> Tiliacoreae

48: Italics for Eleutharrhena

48: the relationship with sister tribes of Tiliacoreae -> the sister relationships of Tiliacoreae

50: and Chen [2] whereas -> and Chen [2], whereas

51: Lian, et al. [5]. -> Lian et al. [5].

51 close parenthesis

42: italics for Macrococculus and Eleutharrhena

58: italics for Eleutharrhena and Pycnarrhena

58: their range overlap -> their ranges overlap

64: Hou et al.[9] -> Hou et al. [9] [insert space]

65: Lang et al.[1] -> Lang et al. [1] [insert space]

76: tertiary -> Tertiary

79: later?

87: italics for Eleutharrhena

100: italics for Simonenium acutum

101: italics for Stephania yunnanensis

102: italics Chasmanthera dependens

104: phylogenetic what? phylogenetic  is an adjective, the noun is lacking

107: How the genetic -> Specifically, we investigate how the genetic

111: Tiliacoreae as well as providing -> Tiliacoreae. Finally, provide data on

121: Wefferling, et al. [20]. -> Wefferling et al. [20].

126: [2,5,21], (see -> [2,5,21] (see

131: analysis,-> analysis;

156: italics for Triclisia inflata

171: Falush, et al. [35] -> Falush et al. [35]

169: the Nei's genetic -> Nei's genetic [for consistence with lines 168, 173, etc]

183: Mantel test -> Mantel’s test

191: in H. validus whereas -> in H. validus, whereas

200: Italics for Macrococculus

202: italics for Eleutharrhena

Figure 2: use capitals for Ventral ridge, Cotyledon, Dorsal ridge, for consistence with the other labels

211-215: italics for all species names

223-226: italics for all species names

Figure 3, inset: umber -> Number

Figure 3: normal, not italics, for XSBN Song, XSNB Zhou…, XSBN Shen and Dehong. Only species names must be in italics

235: not italics for sp.

243: Syrrheonema

245: italics for Pleogyne and Carronia

247: italics for Albertisia and Tiliacora

248-250: italics for Macrococculus, Eleutharrhena and Tiliacora

256: index H, ranged from 0.1334 to 0.3120 and the Shannon’s information index  -> index H ranged from 0.1334 to 0.3120, and Shannon’s information index I

270: are all in subpopulation 2 and subpopulations 2 and -> are all in subpopulation 2, and subpopulations 2 and [correct?]

273: subpopulation 2 and 3. -> subpopulations 2 and 3.

276: subpopulations 2 and 3 -> subpopulations 2 and 3

Fig. 4: use larger font for region names, as they are difficult to read

286: Mantel test -> Mantel’s test

303: embryos (Figure 5), [2]. -> embryos (Figure 5) [2].

309: italics for Macrococculus and Eleutharrhena

313-320: italics fro all species names

Figure 5: normal, not italics, for XSBN Song, XSNB Zhou…, XSBN Shen and Dehong. Only species names must be in italics

328: Phylogenetic -> phylogenetic

329-338: italics for all species names

334: Chen [2], (PP -> Chen [2]

337: share -> shares

341: the tribe much more recent history c. 34.21 Mya, indicates the tribe -> the much more recent history of the tribe (c. 34.21 Mya) indicates that it

343: old world -> Old World

345-347: italics for all species names

351: italics for Pleogyne and Carronia

357-362: italics for all species names

376: within pop 83.62% Among pop 16.38% Nm=2.552), [17]. -> within pop 83.62%, among pop 16.38%; Nm=2.552) [17].

380: Hardy Weinberg equilibrium), [55]. -> Hardy-Weinberg equilibrium) [55].

390: cpDNA, FST=0.797 and low population differentiation and strong geographical correlation for AFLP, FST =0.064 -> cpDNA, with FST=0.797, and low population differentiation and strong geographical correlation for AFLP, with FST =0.064

393: I=0.43401), [18]. -> I=0.43401) [18].

406: The Hua line, -> the Hua line,

410: than the -> with respect to [correct?]

415-419: italics for all species names

425-426: italics for all species names

436: italics for Syrrheonema

440-444: italics for all species names

448: italics Eleutharrhena

Appendix A: only species names in italics

486: Mantel test between genetic distance and geographical distance

Author Response

Thanks very much for giving us a chance to improve our paper. We carefully changed according to your suggestions.

Analyses are based on already published sequences, but one, for which the authors provide GenBank accession code (If I have correctly read; please, confirm).

Yes, this was not very clear: One additional sample collected in Xishuangbanna was extracted from the whole chloroplast assembly of Eleutharrhena macrocarpa (GenBank:MZ502223).

So, we changed to: One additional DNA sample collected in Xishuangbanna was extracted and its whole chloroplast was sequenced using NGS, aligned, assembled and annotated by us (GenBank:MZ502223, detailed methods not shown here). The regions of interest were then extracted using Geneious and added to our matrix. 

I have no particular concern about the methods, results and interpretations.

However, the language must be carefully revised. I’m surprised that the text is full of errors/typos, given the presence of a native speaker among the co-authors. Many sentences are obscure.

Sorry I am not English; the English grammar and typos were corrected and the manuscript was sent for professional English checking. 

Specific comments

86-94: A very messy sentence. Rephrase

Changed to: The floras of southern and southeastern Yunnan have a high proportion of tropical Asian elements, but the southern part is more closely related to the Indo-Malaysian flora suggesting a boundary exists between these two parts [14]. Eleutharrhena is a typical Indo-Malaysian element and may closely correspond to the biogeographical events that lead to the formation of the southern Yunnan flora. Several biogeographical lines have been proposed to explain floral boundaries in Yunnan; the Hua line between southern and southeastern Yunnan floras, the Tanaka line [15] was used to describe the difference between the Sino Himalayan forests to the west and the Sino Japanese forests to the east and the Salween-Mekong divide separating the East Himalayan from the Hengduan mountains floras during glacial periods [16].

95: possibly the authors mean: may not all apply to the tropical flora of southern Yunnan or may be valid only for a part of it.

Change to: However, the influence of these different biogeographical lines and the period of occurrence may not all apply to the tropical flora of southern Yunnan or may only be valid for a part of it.

80-81: add references to support this statement

Several distinct floristic regions and biogeographical areas can be recognized in tropical southern China; Taiwan, Hainan, part of Guangxi and southeastern Yunnan as well as southern Yunnan.

This is from the same reference as before we added [13]

86 between them is not clear, rephrase

Changed to: between these two parts

123: You did not download species, but genetic sequences. Please, correct

62 species in Menispermaceae as well as all available species in tribe Tiliacoreae were downloaded from National Center for Biotechnology Information for the phylogenetic study

Changed to: 62 genetic sequences in Menispermaceae, representing all available species in tribe Tiliacoreae were downloaded from National Center for Biotechnology Information for the phylogenetic study

Figure 3: explain colours in pie charts

The colours in the pie are the same as on the small insert map. We added description in caption for slices with more than 50%: red: South America; blue: Africa; yellow: Australasia; dark green: Indo-China; light green: Aus-tralasia + Indochina

260-61: this sentence is confused, please revise

The mean genetic diversity value is low(H=0.2577, I=0.3905), gene flow is medium [43] Nm=0.6983 and GST (among populations) = 0.4173 showing a high population differentiation [44]

Changed to: The mean genetic diversity value (H=0.2577, I=0.3905) is low, gene flow (Nm=0.6983) is medium following the standard used in [43] and the coefficient of gene differentiation GST (among populations) = 0.4173; all these values show a high population differentiation [44].

304-305: this sentence is confused, please revise

As in other Menispermaceae tribes, the lack of morphological data resulting from the dioecious nature of the family as well as few fruiting specimens, result in patchy morphological data analyses.

Changed to: Lack of morphological data especially in flowers of both sexes and fruits have resulted in patchy morphological data analyses in Menispermaceae tribes.

307-308: this sentence is confused, please revise

No synapomorphies in sepal number and number of whorls could be identified using the synopsis of the family [3], however hippocrepiform seeds, without endosperm and a conspicuous condyle seem to be a symplesiomorphy of the tribe with a distinct reversion in Macrococculus and Eleutharrhena where the condyle has completely disappeared and instead an aril like tissue is present (Figure 2).

Using the morphological characters listed in the synopsis of the family [3], the number of sepals and number of perianth whorls were not shared derived characters among the genera of tribe Tiliacoreae, however hippocrepiform seeds, without endosperm and a conspicuous condyle seem to be a shared primitive character of the tribe expect within Macrococculus and Eleutharrhena. In these two genera the condyle has completely disappeared and instead an aril like tissue is present (Figure 2).

309: does “where” refer to both genera, or only to Eleutharrhena. Please, revise to avoid ambiguity

Changed to: In these two genera the condyle has completely disappeared and instead an aril like tissue is present (Figure 2).

Figure 5: explain the meaning of perf and no. Explain the meaning of the numbers and the scale bare.

Perf: perforate, no: absent. The scale bar was deleted as it was not deemed relevant in this figure. (it represents the age scale in Figtree)

358-9: confused; use (1) … (2) …(3)…. to clarify

We added (1) (2) and (3) for the three groups within the tribe and sub consequently used the same numbers in the rest of the text

365-368: revise punctuation or divide this long sentence into smaller ones

A relatively high number of bands was obtained from the SCoT analysis although between 53712 and 78921 unigenes were detected in Menispermum canadense and M. dauricum [55], our total bands therefore represent 0.7% to 1% of the total number of unigenes, this may be due to the use of fluorescent primers and capillary electrophoresis.

Change to: A relatively high number of bands was obtained from the SCoT analysis although between 53712 and 78921 unigenes were detected in Menispermum canadense and M. dauricum [55]. Our total bands therefore represents 0.7% to 1% of the total number of unigenes, this may be due to the use of fluorescent primers and capillary electrophoresis.

399-404:  confused, please revise (for example, it is not clear whether “which result” refers to specific habitat requirement; in this case result -> results). A full revision of this sentence is needed

We added (1) and (2) to show the two types of opposite forces that could lead to the low level of genetic diversity and medium gene flow among populations but high population differentiation. These two results are contradictory why we propose these two opposite forces. This part is a discussion about these contradictory results.

However, two populations in Dehong and Honghe show relatively higher genetic diversity and population differentiation. Low level of genetic diversity and medium gene flow among populations but high population differentiation might be due to mixed forces involving small population sizes, specific habitat requirement which result in lower within population diversity, with sympatric barriers such as mountain ranges or rivers, pollinators and short seed dispersal distances which result in higher population differentiation [58]

Changed to: However, two populations in Dehong and Honghe show relatively higher genetic diversity and population differentiation. Low level of genetic diversity and medium gene flow among populations but high population differentiation might be due to mixed forces involving (1) small population sizes and specific habitat requirement which result in lower within population diversity, with (2) sympatric barriers such as mountain ranges or rivers, pollinators specificity and short seed dispersal distances which result in higher population differentiation [58]

406: provide references for the Hua line, the Tanaka line and the Salween Mekong divide

Hua line [14], the Tanaka line [15] and the Salween Mekong divide [16]

415-19: messy sentence; revise, please

Finally, Eleutharrhena as well as Macrococculus have high habitat requirement which may be reflected by their disjunct distribution, the development of the monsoon climate and the Himalayan uplift for Eleutharrhena and contribute to the higher population differentiation despite a low genetic diversity.

Changed to: Finally, Eleutharrhena has a high habitat requirement and can only be found along small streams within primary forests between 800 to 1500m high (cite eFlora of china). This may have contributed to the higher population differentiation despite a low genetic diversity. We can infer that this specificity is also linked to the Himalayan uplift and the development of a monsoon climate in southern Yunnan, whereas a similar process may have occurred in the sister and disjunct genus Macrococculus in New Guinea.

425-430: messy sentence; revise, please

Our results show that Eleutharrhena has a high phylogenetic distinctiveness value because it is related to Macrococculus and their unique aril like structure surrounding the seed, biogeographic value as shown by the disjunction between these two genera as well as the isolation of the Dehong population and as a representative of the migration and distribution of plants within SE Asia rainforests from the Pleistocene to now

Changed to: Our results show that Eleutharrhena has a high phylogenetic distinctiveness value because it is a monospecific genus sister to the more diversified Pycnarrhena genus. It is also more closely related to the genus Macrococculus using several morphological characters including the unique aril like structure surrounding the seed. These two genera are monospecific, disjunct across the Wallace line and occur in similar habitat adding to the valuable biogeographic history of clade (2). The evolutionary history of tribe Tilacoreae from South America, Africa and finally SE Asia is different from previously studied tribes within Menispermaceae and it is the first time we document the differentiation across the Thai–Burmese border range for any plant species in southern Yunnan. E. macrocarpa is a representative species for the migration and distribution of plants with SE Asia most northerly rainforest from the Pleistocene to now.

430-432: something is lacking. That have colonized what? When?

Yes it is not clear the hypothesis is the genus was more widespread in SE Asia and then migrated towards its northern range after the Himalayan uplift.

Eleutharrhena macrocarpa is a relict plant of the mighty rainforests that have colonized and persisted in the northern and southern range of the southeast Asian biodiversity hotspot.

Changed to: Eleutharrhena macrocarpa is a relict plant that has colonized and persisted in southeast Asia most northerly rainforests following the Himalayan uplift and the apparition of more suitable habitats.

Typos/grammar

The manuscript is full of typos.

The tribe name Tiliacoreae is misspelled as in various parts, even in the title.

 Changed throughout

Just a few examples of typos/grammar issues Tiliacorea

42: Tiliacorea -> Tiliacoreae

changed

44: delete comma after  [1]

changed

45: tiliacoreae -> Tiliacoreae

changed

48: Italics for Eleutharrhena

All genera names were changed to italics throughout

48: the relationship with sister tribes of Tiliacoreae -> the sister relationships of Tiliacoreae

changed

50: and Chen [2] whereas -> and Chen [2], whereas

changed

51: Lian, et al. [5]. -> Lian et al. [5].

changed

51 close parenthesis

added

42: italics for Macrococculus and Eleutharrhena

changed

58: italics for Eleutharrhena and Pycnarrhena

changed

58: their range overlap -> their ranges overlap

changed

64: Hou et al.[9] -> Hou et al. [9] [insert space]

changed

65: Lang et al.[1] -> Lang et al. [1] [insert space]

changed

76: tertiary -> Tertiary

changed

79: later?

We did not find this problem

87: italics for Eleutharrhena

changed

100: italics for Simonenium acutum

changed

101: italics for Stephania yunnanensis

changed

102: italics Chasmanthera dependens

changed

104: phylogenetic what? phylogenetic  is an adjective, the noun is lacking

Changed to phylogenetics

107: How the genetic -> Specifically, we investigate how the genetic

Changed

111: Tiliacoreae as well as providing -> Tiliacoreae. Finally, provide data on

Changed

121: Wefferling, et al. [20]. -> Wefferling et al. [20].

Changed

126: [2,5,21], (see -> [2,5,21] (see

Changed

131: analysis,-> analysis;

Changed

156: italics for Triclisia inflata

Changed

171: Falush, et al. [35] -> Falush et al. [35]

Changed

169: the Nei's genetic -> Nei's genetic [for consistence with lines 168, 173, etc]

Changed

183: Mantel test -> Mantel’s test

Changed

191: in H. validus whereas -> in H. validus, whereas

Added

200: Italics for Macrococculus

Changed

202: italics for Eleutharrhena

Changed

Figure 2: use capitals for Ventral ridge, Cotyledon, Dorsal ridge, for consistence with the other labels

Changed

211-215: italics for all species names

Changed

223-226: italics for all species names

Changed

Figure 3, inset: umber -> Number

Changed

Figure 3: normal, not italics, for XSBN Song, XSNB Zhou…, XSBN Shen and Dehong. Only species names must be in italics

Changed

235: not italics for sp.

Changed

243: Syrrheonema

removed

245: italics for Pleogyne and Carronia

Changed

247: italics for Albertisia and Tiliacora

Changed

248-250: italics for Macrococculus, Eleutharrhena and Tiliacora

Changed

256: index H, ranged from 0.1334 to 0.3120 and the Shannon’s information index  -> index H ranged from 0.1334 to 0.3120, and Shannon’s information index I

Changed

270: are all in subpopulation 2 and subpopulations 2 and -> are all in subpopulation 2, and subpopulations 2 and [correct?]

Changed to: Subpopulation 1 has 2 individuals from Dehong. Individuals from Lincang are all in subpopulation 2. Subpopulations 2 and 3 are mixed between Xishuangbanna, Dehong, Honghe, Lincang and Pu’er individuals

273: subpopulation 2 and 3. -> subpopulations 2 and 3.

Changed

276: subpopulations 2 and 3 -> subpopulations 2 and 3

Changed

Fig. 4: use larger font for region names, as they are difficult to read

Changed

286: Mantel test -> Mantel’s test

Changed

303: embryos (Figure 5), [2]. -> embryos (Figure 5) [2].

Changed

309: italics for Macrococculus and Eleutharrhena

Changed

313-320: italics fro all species names

Changed

Figure 3: normal, not italics, for XSBN Song, XSNB Zhou…, XSBN Shen and Dehong. Only species names must be in italics

Changed

328: Phylogenetic -> phylogenetic

Changed

329-338: italics for all species names

Changed

334: Chen [2], (PP -> Chen [2]

Changed

337: share -> shares

Changed

341: the tribe much more recent history c. 34.21 Mya, indicates the tribe -> the much more recent history of the tribe (c. 34.21 Mya) indicates that it

Changed

343: old world -> Old World

Changed

345-347: italics for all species names

Changed

351: italics for Pleogyne and Carronia

Changed

357-362: italics for all species names

Changed

376: within pop 83.62% Among pop 16.38% Nm=2.552), [17]. -> within pop 83.62%, among pop 16.38%; Nm=2.552) [17].

Changed

380: Hardy Weinberg equilibrium), [55]. -> Hardy-Weinberg equilibrium) [55].

Changed

390: cpDNA, FST=0.797 and low population differentiation and strong geographical correlation for AFLP, FST =0.064 -> cpDNA, with FST=0.797, and low population differentiation and strong geographical correlation for AFLP, with FST =0.064

Changed

393: I=0.43401), [18]. -> I=0.43401) [18].

Changed

406: The Hua line, -> the Hua line,

Changed

410: than the -> with respect to [correct?]

Changed to: It is therefore possible that  Dehong  may represent a separate refugium from  the rest of the populations distribution

415-419: italics for all species names

Changed

425-426: italics for all species names

Changed

436: italics for Syrrheonema

Changed

440-444: italics for all species names

Changed

448: italics Eleutharrhena

Changed

Appendix A: only species names in italics

Changed

486: Mantel test between genetic distance and geographical distance

Changed to Mantel’s test

Reviewer 2 Report

The present study by Song et al. reported the “Biogeography and population history of Eleutharrhena macrocarpa (Tiliacorea tribe) in Southeast Asia most northerly rainforests.” The work is interesting and thoroughly documented, for this reason, I recommend the acceptance of the manuscript. However, in my opinion, the authors could accompany their phylogeny results with better storytelling.

Hereafter, some issues arisen during reading:

Please check: the complete scientific name and authority should be given for every organism the first time it is mentioned.

I would suggest checking and changing all the words that are needed to be written in italics (e.g. species).

Keywords: I suggest replacing the name of the genera that are also in the title and in the abstract with other words (e.g. Eleutharrhena with Menispermaceae).

Pag. 3 line 91: the verb describe shouldn’t be in past tense

Pag. 4 line 119: could authors clarify why they compared the specimen fruit of Eleutharrhena macrocarpa with that of Haematocarpus Validus? Is it a phylogenetically related genus?

Pag. 5, paragraph 2.4 “Population genetic analysis”, please check the interline spacing

Pag. 9, paragraph 3.5 “Population genetic diversity”, please check the interline spacing

Pag. 15, paragraph 4.4 “Evolutionary distinctiveness”, please check the interline spacing

Author Response

Thanks very much for giving us the chance to improve the quality of our paper. We changed all the mistakes in our manuscript according to your valuable suggestions.

Please check: the complete scientific name and authority should be given for every organism the first time it is mentioned.

Changed

I would suggest checking and changing all the words that are needed to be written in italics (e.g. species)

Sorry we had copy paste problem there are many genera and species name not in italics now we have changed all of them

Keywords: I suggest replacing the name of the genera that are also in the title and in the abstract with other words (e.g. Eleutharrhena with Menispermaceae).

Added Wallace line and seed morphology instead of these genera names

Pag. 3 line 91: the verb describe shouldn’t be in past tense

changed

Pag. 4 line 119: could authors clarify why they compared the specimen fruit of Eleutharrhena macrocarpa with that of Haematocarpus Validus? Is it a phylogenetically related genus?

Thanks this was not clear. Haematocarpus is often misidentified as Eleutharrhena in its vegetative state, and it used to be recognised within tribe Tiliacoraea based on its seed morphology. It was chosen to compare with a deeply curved seed with conspicuous condyle which is characteristic of the tribe Tiliacoreae.

Added: For comparison with seeds having a strongly curved embryo, a conspicuous condyle and intrusive tissue such as the raphe we also dissected for the first time, the fresh fruits of Haematocarpus validus (tribe Pachygoneae). Haematocarpus is often misidentified with E. macrocarpa in the field but markedly differ in its seed morphology.

Pag. 5, paragraph 2.4 “Population genetic analysis”, please check the interline spacing

changed

Pag. 9, paragraph 3.5 “Population genetic diversity”, please check the interline spacing

changed

Pag. 15, paragraph 4.4 “Evolutionary distinctiveness”, please check the interline spacing

changed

Reviewer 3 Report

With interest I have read the manuscript, “Biogeography and population history of Eleutharrhena macrocarpa (Tiliacorea tribe) in Southeast Asia most northerly rainforests”. The authors sampled 48 individuals of Eleutharrhena macrocarpa to discuss the population history. They proposed E. macrocarpa was once distributed across the Wallace line and is a relict species restricted in the southern Yunnan rainforests. Besides, they pointed the Thai–Burmese range might have played a major role in the isolation of the Dehong populations. However, the scope and the results of this paper are not correspond, and the main conclusions are unconvincing. There are several major points that I think would need attention.

  1. Based on the endocarp morphology, the authors suggest a close relationship of Macrococculus and Eleutharrhena. Importantly, one of the main conclusion of the paper is based on it. However, the evidences are very insufficient because inconspicuous condyle is found in many other genera of Menispermaceae, such as: Pycnarrhena and Arcangelisia. The Macrococculus is not sampled in both the molecular phylogenetic analysis and endocarp observation. For a convincing result, I strongly recommend the authors to sample it. Otherwise, the discussion of biogeography history of Macrococculus should be removed.

  1. For the part of morphological observation, only two endocarps of Eleutharrhena and Haematocarpus validus (a species from the other tribe, Pachygoneae) were dissected and observed. Why Haematocarpus validus from the other tribe is observed in this study? Why don’t you choose species who has a close relationship to Eleutharrhena (like Pycnarrhena or Macrococculus)?

  1. The biogeography history of the tribe is far beyond the scope of this paper. With one new sequenced Eleutharrhena macrocarpa, the authors reconstructed the biogeographic history of Tiliacoreae, which is unreasonable. Instead of the ancestral area reconstruction of Tiliacoreae, the authors should focus on the biogeographic history of different populations of Eleutharrhena to match the title and main scope of the paper. And the dating result of the tribe should be used as second calibration points for investigating the divergence time of populations.

  1. No description of ancestral area reconstruction was find in the “2.3 Phylogenetic analysis, dating and ancestral area reconstruction”, which is totally incomprehensible.

  1. For the population level, which genes are sequenced should be clarified. Besides, a dating analysis at population level is needed to investigate the population history.

  1. In the result of Dating analyses, the authors used the totally wrong chronostratigraphy (Figure 3). Thus, all the diverging times were attributed to the wrong stratum. For example, 34.21 Mya (Line 220) is in Eocene, not Miocene. And this time is not located in the following 95% HPD: 42.72-84.56 Mya. Right application of chronostratigraphy is the basis of biogeography discussion. Due to the basic concept error, I have to question the reliability of the subsequent discussion and conclusions.

  1. One of the main conclusion of the paper is “the Thai–Burmese range might have played a major role in the isolation of the Dehong populations”, while no Thai–Burmese range was referred in the discussion.

Other minor points:

  1. In the title and key words, the “Tiliacoreae” is misspelled. Please check it through the text.
  2. In the Figure A1, the tribe Pachygoneae and Spirospermeae were mislabeled. The PP and BS values on the tree are contrary to the figure legend.
  3. Line 35-37: This sentence need to be rephrased.
  4. L45: “Tiliacoreae”, not “tiliacoreae”.
  5. L47: there are 111 species of Tiliacoreae in Ortiz et al. (2016), not 95.
  6. L51: seven plastid regions were used in Lian et al. (2021), not five.
  7. L69-75: there are no directly relevant to this paper.
  8. L76: “Tertiary”, not “tertiary”.
  9. L80-84: please provide the references.
  10. L87-89: please provide the references.
  11. L90: please provide reference to the “Hua line”.
  12. L106: no questions about the phylogeography of tribe Tiliacoreae is answered in this paper.
  13. L112-113: “save evolutionary history” is illogical.
  14. L126: Lian et al. (2021) sampled 4 individuals of Eleutharrhena macrocarpa, but only three individuals are included in this study. For a more comprehensive sampling, as many as possible individuals of Eleutharrhena should be included.
  15. L132: 500m is not an effective distance for separating population. And for Figure 1, the authors should focus on the Eleutharrhena populations in Yunnan.
  16. L220: 34.21 Mya is in Eocene, not Miocene. And this time is not located in the following 95% HPD: 42.72-84.56 Mya.
  17. L222: 23.46 Mya is in Oligocene, not Pliocene.
  18. L225: 27.13, 27.47 and 28.15 Mya are in Oligocene, not late Pliocene.
  19. L227: 17.65 Mya is in Miocene, not Pleistocene.
  20. L229: 14.39 Mya is in Miocene, not Pleistocene.
  21. L231: “PP = 0.96”, not “PP=0,.96”.
  22. Figure 3: fossil “Diplocisia bognorensis 48.6Mya” is in a relative young placement, which is obvious incorrect. Strychnopsis thouarsii is a member of Spirospermeae, not Cissampelideae. The chronostratigraphy is totally wrong.
  23. L268: no description of “Principal Coordinate Analysis” in the Material and Methods part.
  24. L293-300: these should be the content of Introduction.
  25. L336-337: “Pleiogyne” is wrong spelled. And the phylogenetic relationship should not be discussed based on an incongruent result.
  26. L341-343: in the phylogenetic result, Tiliacoreae and other three tribes are in an unresolved relationship. It is irresponsible to infer biogeography history based on these.
  27. L339-357: these discussion are far beyond the scope of this paper.
  28. L415-419: no Macrococculus population were sampled.

Author Response

With interest I have read the manuscript, “Biogeography and population history of Eleutharrhena macrocarpa (Tiliacorea tribe) in Southeast Asia most northerly rainforests”. The authors sampled 48 individuals of Eleutharrhena macrocarpa to discuss the population history. They proposed E. macrocarpa was once distributed across the Wallace line and is a relict species restricted in the southern Yunnan rainforests. Besides, they pointed the Thai–Burmese range might have played a major role in the isolation of the Dehong populations. However, the scope and the results of this paper are not correspond, and the main conclusions are unconvincing. There are several major points that I think would need attention.

Many thanks for your very knowledgeable review on the biogeography part we think you have pointed out the weaknesses of our article very well. We followed most of your comments and reduced the biogeography to focus only on Eleutharrhena and Pycnarrhena. We kept the tribe phylogeny only for background and discussion, but we believe the morphological description of Macrococculus and Eleutharrhena seeds gives strong evidence of their sister relationship. We only incorporated this relationship within the morphology part and not the phylogeny and biogeography part as DNA samples of this genus are not available.  

Many thanks for suggesting using the plastid regions of E. macrocarpa for the biogeography and dating analysis unfortunately resolution within species using such methods would not be satisfactory. Most studies using this method are at least within genus level or use the haplotype data to infer population structure. This is why we also incorporated Pycnarrhena within the analysis. 

  1. Based on the endocarp morphology, the authors suggest a close relationship of Macrococculus and Eleutharrhena. Importantly, one of the main conclusion of the paper is based on it. However, the evidences are very insufficient because inconspicuous condyle is found in many other genera of Menispermaceae, such as: Pycnarrhena and Arcangelisia. The Macrococculus is not sampled in both the molecular phylogenetic analysis and endocarp observation. For a convincing result, I strongly recommend the authors to sample it. Otherwise, the discussion of biogeography history of Macrococculus should be removed.

Thanks, the key character is the aril like structure which is shared between Eleutharrhena and Macrococculus not the condyle. Besides all other morphological characters such as sepal number and whorls number are congruent with this close relationship rather than between Eleutharrhena and Pycnarrhena. We removed all conclusions based on phylogenetic and biogeography results to only incorporate this part in discussion and results about seed morphology.

 For the part of morphological observation, only two endocarps of Eleutharrhena and Haematocarpus validus (a species from the other tribe, Pachygoneae) were dissected and observed. Why Haematocarpus validus from the other tribe is observed in this study? Why don’t you choose species who has a close relationship to Eleutharrhena (like Pycnarrhena or Macrococculus)?

 We chose a species that had opposite characters from Eleutharrhena, Pycnarrhena and Macrococculus for comparison purposes. It would have been good to have Pycnarrhena and Macrococculus seeds as well but we do not have access to these, and several species were previously described in other publications, so we think the necessary material is presented to answer our question. Although further study at species level within Pycnarrhena would benefit from more seeds being dissected.

Added: For comparison with seeds having a strongly curved embryo, a conspicuous condyle and intrusive tissue such as the raphe we also dissected for the first time, the fresh fruits of Haematocarpus validus (tribe Pachygoneae). Haematocarpus is often misidentified with E. macrocarpa in the field but markedly differs in its seed morphology

  1. The biogeography history of the tribe is far beyond the scope of this paper. With one new sequenced Eleutharrhena macrocarpa, the authors reconstructed the biogeographic history of Tiliacoreae, which is unreasonable. Instead of the ancestral area reconstruction of Tiliacoreae, the authors should focus on the biogeographic history of different populations of Eleutharrhena to match the title and main scope of the paper. And the dating result of the tribe should be used as second calibration points for investigating the divergence time of populations.

Thanks we understand your concern and reduced the scope to only include Eleutharrhena and Pycnarrhena. For the population genetic part with used SCOT method which we have not seen used to reconstruct phylogenetic tree and dating analysis.

  1. No description of ancestral area reconstruction was find in the “2.3 Phylogenetic analysis, dating and ancestral area reconstruction”, which is totally incomprehensible.

 Our apologies this part was missing in this last edition we now added it back

Ancestral ranges of Menispermaceae tribe Tiliacoreae and Eleutharrhena clade were inferred with the package ‘BioGeoBEARS’ [32] implemented in RStudio [33]. Tribe Tiliacoreae genera were extracted from the maximum clade credibility tree inferred with BEAST, and input in BioGeoBEARS. Six models were tested DEC, DEC+J, DI-VALIKE, DIVALIKE+J, BAYAREALIKE and BAYAREALIKE+J[32] and the most fit-ting model was selected by calculating the best value for the Likelihood Ratio Test (LRT). Four major geographical ranges were designed: Indochina, Southern Yunnan, Sundaland including Wallacea and Australasia including New Guinea. The resulting probabilities of the ancestral states were drawn as pie charts at the node on the provided tree.

  1. For the population level, which genes are sequenced should be clarified. Besides, a dating analysis at population level is needed to investigate the population history.

Thanks for this suggestion our decision was not to use plastid regions for the dating and studying the population history of E. macrocarpa. Few studies have used plastid regions to do dating at population level probably due to the nature of hybridisation and more complex history that cannot be shown using plastid regions. Although some studies use them to study haplotypes as done in this publication similar to ours (Iloh AC, Schmidt M, Muellner-Riehl AN, Ogundipe OT, Paule J (2017) Pleistocene refugia and genetic diversity patterns in West Africa: Insights from the liana Chasmanthera dependens (Menispermaceae). PLoS ONE 12(3): e0170511. https://doi.org/10.1371/journal.pone.0170511) we did not use them as we only have 4 individuals for this study and the SCoT analysis based on many nuclear markers is more reliable.

  1. In the result of Dating analyses, the authors used the totally wrong chronostratigraphy (Figure 3). Thus, all the diverging times were attributed to the wrong stratum. For example, 34.21 Mya (Line 220) is in Eocene, not Miocene. And this time is not located in the following 95% HPD: 42.72-84.56 Mya. Right application of chronostratigraphy is the basis of biogeography discussion. Due to the basic concept error, I have to question the reliability of the subsequent discussion and conclusions.

Thanks for pointing this out it is a big mistake we changed. We used a non to scale chronostratigraphy. The discussion on biogeography was based on the results from the Beast analysis so we did not use the figure with wrong chronostratigraphy. We doubled checked to make sure the discussion and conclusion correspond to the beast results.

We also checked all over dates and HPD on our original beast tree in figtree

For the new version we redid the analysis so we used new dates HPD and chronostratigraphy

One of the main conclusions of the paper is “the Thai–Burmese range might have played a major role in the isolation of the Dehong populations”, while no Thai–Burmese range was referred in the discussion.

The thai-Burmese range is discussed in paragraph 4.4, 4.3 and also mentioned in abstract and conclusion.

However, differentiation across the Thai–Burmese border range (Tenasserim Hills, Dawna range, and Karen Hills) was proposed to explain the diversity in several bird and mammal species [59].

This hypothesis is however not certain as this is the first time this was suggested in any biogeography pattern for plants. Work is still lacking between southern China and Burma/India plant distribution.

Other minor points:

  1. In the title and key words, the “Tiliacoreae” is misspelled. Please check it through the text.

Changed and checked throughout

  1. In the Figure A1, the tribe Pachygoneae and Spirospermeae were mislabeled. The PP and BS values on the tree are contrary to the figure legend.
  2. Line 35-37: This sentence need to be rephrased.

Populations with high genetic diversity and unique evolutionary history such as Eleutharrhena, Macrococculus and from Dehong should be prioritized for conservation. 

Changed to: Dehong populations with high genetic diversity should be prioritized for conservation and the unique evolutionary history of Eleutharrhena should be considered to assess its conservation status. 

  1. L45: “Tiliacoreae”, not “tiliacoreae”.

Changed

  1. L47: there are 111 species of Tiliacoreae in Ortiz et al. (2016), not 95.

The references we cite list much less species but they have been published previously so we changed to 111. Thanks.

  1. L51: seven plastid regions were used in Lian et al. (2021), not five.

Thanks this was changed

  1. L69-75: there are no directly relevant to this paper.

Although it has no more direct reference to our study we used these papers a lot for inspiration so we think they are necessary to understand the context about Menispermaceae biogeography.

  1. L76: “Tertiary”, not “tertiary”.

Changed

  1. L80-84: please provide the references.

Added [13]

  1. L87-89: please provide the references.

This sentence is our hypothesis

  1. L90: please provide reference to the “Hua line”.

Added [14]

  1. L106: no questions about the phylogeography of tribe Tiliacoreae is answered in this paper.

The question is to reconstruct the phylogeography of tribe Tiliacoreae and understand the evolutionary history of Eleutharrhena within this tribe.

This part is not relevant anymore in the new version

  1. L112-113: “save evolutionary history” is illogical.

The idea was to show that if E. macrocarpa becomes extinct, we also lose access to its evolutionary history and to show it evolutionary history is unique so it should get conservation priority

This was deleted in latest version

  1. L126: Lian et al. (2021) sampled 4 individuals of Eleutharrhena macrocarpa, but only three individuals are included in this study. For a more comprehensive sampling, as many as possible individuals of Eleutharrhena should be included.

The sample Zhou HS 8508 (Jiangcheng) was not added because most of the sequences were missing in genebank and only partial. Only around 15% of the sequences is available for this sample.

  1. L132: 500m is not an effective distance for separating population. And for Figure 1, the authors should focus on the Eleutharrhena populations in Yunnan.

Yes, this was chosen arbitrarily as we could not find any reference for effective distance. The closest populations are p10 and p11 which are 4.3 km apart, so we changed to 4 km.

  1. L220: 34.21 Mya is in Eocene, not Miocene. And this time is not located in the following 95% HPD: 42.72-84.56 Mya.

Tribe Tiliacoreae diverged during the Eocene c. 34.21 Mya (18.98-49.10 Mya, 95% Highest Posterior Density interval (HPD))

Changed in new version because on new analysis

  1. L222: 23.46 Mya is in Oligocene, not Pliocene.

Changed very sorry

Changed in new version because on new analysis

  1. L225: 27.13, 27.47 and 28.15 Mya are in Oligocene, not late Pliocene.

Changed very sorry

Changed in new version because on new analysis

  1. L227: 17.65 Mya is in Miocene, not Pleistocene.

Changed very sorry

Changed in new version because on new analysis

  1. L229: 14.39 Mya is in Miocene, not Pleistocene.

Changed very sorry

Changed in new version because on new analysis

  1. L231: “PP = 0.96”, not “PP=0,.96”.

Changed

  1. Figure 3: fossil “Diplocisia bognorensis 48.6Mya” is in a relative young placement, which is obvious incorrect. Strychnopsis thouarsii is a member of Spirospermeae, not Cissampelideae. The chronostratigraphy is totally wrong.

We moved the fossils in the figure to correspond to the dates. These dates were entered in Beauti  and then Beast correctly.

Changed in new version because on new analysis we only chose one fossil.

  1. L268: no description of “Principal Coordinate Analysis” in the Material and Methods part.

We have not seen description of PCoA in other paper on population genetics. Maybe it is not so relevant at this level because this method is widely used across many fields.

  1. L293-300: these should be the content of Introduction.

Moved to introduction

  1. L336-337: “Pleiogyne” is wrong spelled. And the phylogenetic relationship should not be discussed based on an incongruent result.

Corrected. This part was deleted

  1. L341-343: in the phylogenetic result, Tiliacoreae and other three tribes are in an unresolved relationship. It is irresponsible to infer biogeography history based on these.

The support is low but not unresolved. We deleted the two samples with less data Pleogyne and Syrrheonema which gave high support to all branches.

In the new version we did a new analysis this part is not relevant anymore.

  1. L339-357: these discussion are far beyond the scope of this paper.

We removed the discussion about Pleogyne and Syrrheonema only focusing on the tribe, Eleutharrhena, Macrococculus and Pycnarrhena

  1. L415-419: no Macrococculus population were sampled

Thanks the discussion about Macrococculus was deleted in this part we only kept parts that were using the morphological data and in discussion.

Round 2

Author Response

Thanks for giving us another opportunity to improve the manuscript, we carefully read comments and answered all major problems except for the writing and non-standard words which we resolved by sending the manuscript to an English language editing service from MDPI.

The description and figure of the phylogenetic analyses are mismatched, so the topology of the phylogeny tree and dating tree, so is the topology of the phylogeny tree and dating tree:

We redid all analyses, and this problem does not occur anymore.

Phylogenetic tree based on the morphological data is not mentioned in the M&M:
This part has been updated and we added morphological analysis using Mesquite based on the Bayesian analysis.

The description and figure of the phylogenetic analyses are totally mismatched. In the “3.2 phylogenetic analyses” part, the authors described there are three groups within Tiliacoreae. While the topology and supports are both mismatched between the description and Figure A1. According to the description in the result part, the second clade is “the Asian and Australasian genera Carronia, Eleutharrhena and Pycnarrhena (BS=100, PP=0,.96)”. But according to Fig. A1, Carronia is clustered with Eleutharrhena, Pycnarrhena, Tiliacora, Albertisia, Anisocycla, Triclisia, and Beirnaertia. Please note that Anisocycla is missed in the author’s description for clade 3.

We added Anisocycla in the text part. We reran the beast analysis including the sample from Jiangcheng (Zhou H.S. 8505) and now both figures, one based on Maximum Likelihood (presented tree) and Beast based on Bayesian analysis now have the same topology.  

This manuscript is focused on the “Biogeography and population history of Eleutharrhena macrocarpa”, but the monophyly of the Eleutharrhena macrocarpa clade is not with strong supports (PP< 0.80 and BS<60% according to Figure A1). And in my previous comments, I suggest the authors to include all the available individuals of Eleutharrhena macrocarpa for a more comprehensive sampling. While the authors indicate that “The sample Zhou HS 8508 (Jiangcheng) was not added because most of the sequences were missing in genebank and only partial. Only around 15% of the sequences is available for this sample.” Curiously, according to Lian et al. (2021) I found four of the five sequences used in this study and the sequences’ length is far beyond 15%. The genbank accession numbers are list here: MW633356, MW633381, MW633392, MW633402. I hope these will be helpful for the authors.

Thanks for your suggestion the 15% length is compared with our aligned matrix, because the sequences may not match exactly with our alignment. When adding the sample, the support increased, so we added this sample and reran analyses. The posterior probability of Eleutharrhena is now slightly higher to 0.8, however the ML and Bayesian analysis support for the Dehong specimen as sister to the rest of Eleutharrhena samples is still below BS = 60% and PP = 0.8. Lian lian (2021) paper from which this accession is from obtained BS = 98% and PP = 1 for Eleutharrhena but based on seven plastids and two nuclear regions. To check for this, we ran a separate analysis based on 7 sequences and 2 nuclear based on our sample and Lian Lian’s.

both nuclear and plastid separately to test for incongruency but there was no sign of hybridisation for the CPG2942 sample from Dehong.

We therefore completely resampled and used a similar sampling to Lian lian as well as the 7 plastid plus two nuclear regions now. They also did not find incongruency between plastid and nuclear so now the support for most clades including Eleutharrhena is high.

The only problem is the sampling from South American clade (1) is now comparatively poorer so the support rate between Curarea and Sciadotenia is now lower.

The topology of the phylogeny tree and dating tree are also mismatched.

Now corrected.

A phylogenetic tree including Macrococculus and Pleogyne was presented in Figure 5. Both these two species are missed in the molecular phylogeny, it seems like the phylogenetic tree in Figure 5 is constructed based on the morphological data only. But I didn’t find any description about this tree in the Materials and Methods part.

We changed this part and added a part where we mapped morphological characters of endocarps on the ML tree (not including Syrrheonema and Pleogyne which don t have enough DNA regions sequenced). The support for Macrococculus as sister to Eleutharrhena is only based on morphological data and we attached the analysis in Appendix also using Mesquite. The conclusion on the Eleutharrhena-Macrococculus relationship was toned down as already previously suggested and future work including Macrococculus molecular sampling could resolve this question in the future was added in conclusion. 

The irregular writing and careless mistakes are listed here:

Many mistakes are indicated when they first occurred, please correct them in the entire manuscript.

  1. “Tiliacoreae” means the tribe, please change “Tiliacoreae tribe” to “Tiliacoreae”.

Changed

  1. Line 24-27: “endocarp morphology” is not an analysis, it should not be in a parallel relations with “phylogenetic analysis, divergence time estimation, ancestral area reconstruction 25 using five chloroplast regions, and SCoT”.

We added the mapping of morphological characters on the phylogenetic tree using Mesquite so we think the morphology part now also works well with the other analyses.

  1. L42: the comma should not appear after “Keywords:” And the keywords should be in alphabetical order. By the way, same font size please.

changed

  1. L48-49: “southern Yunnan” is in a parallel relationship with “China, Laos, Myanmar and India”.

we changed to China (southern Yunnan)

  1. L61-62: I don’t think the phylogeographic studies are reliable. A reliable phylogeographic studies should based on a robust phylogenetic analysis first.

We wanted to explain that without easily identifiable fossils it is not possible to obtain reliable dated phylogenies and good phylogeographic history. So, we changed to dated phylogenies to make it clearer.

  1. Figure 1: Macrococculus is wrong spelled. And what is clade (2)? Please separate the Distribution map of Tiliacoreae and Eleutharrhena as two independent pictures to avoid unnecessary disputes.

Changed Macrococculus spelling.  Clade (2) is Eleutharrhena + Pycnarrhena + Carronia as written in results part. We changed the title of distribution map to Eleutharrhena and Pycnarrhena as well as sister genera Pleogyne and Carronia. We did not add Tiliacora and Albertisia so the map is clearer and less overlap between these genera.

  1. L70: Macrococculus is not sampled in Lian et al. (2021). How did you come to such a conclusion?

Thanks, this was not clear we changed to:

Eleutharrhena is a sister to Pycnarrhena [5] and morphologically similar to Macrococculus [45].

45: Forman, L. The Menispermaceae of Malesia and adjacent areas: VI: Pycnarrhena, Macrococculus & Haematocarpus. Kew Bulletin 1972, 26, 405-422.

  1. L79-88: The authors described the origin of Menispermaceae and some dispersal events in the other tribes, but all these seem no direct connection with this article.

We deleted most of this part only keeping the first sentence about the relation between Menispermaceae and the expansion of rainforests worldwide.

  1. L95-96: The comma should not be here.

deleted

  1. L126-127: the “evolutionary history” can’t be saved from extinction.

We deleted evolutionary history

  1. L135: “the Pachygoneae tribe” change to “Pachygoneae”.

changed

  1. L135-136: please provide the references.

Added Lian Lian [5]

  1. Please unified the format of all the software including with or without “v.”.

Changed and unified throughout

  1. L183-184: Eleutharrhena is included in Tiliacoreae.

We deleted Eleutharrhena in this sentence

15: Figure 2: Please indicate the transverse section and sagittal section for Haematocarpus validus too.

Transverse and sagittal section added for Haematocarpus validus

  1. L243: a lot of strange brackets are found after HPD. Please change all the 95% HPD to this uniform format: 95% HPD: XX-XX Mya.

Removed redundant brackets. Changed all HPD formatting

  1. Figure 3: there is no dispersal event within Eleutharrhena. “Best model DEC+J 4 areas max, d=0, e=0, ej=0.13, LnL=-14.3.” is not a sentence. “Red”, not “red”. By the way, there should be space on both side of Equal sign.

We think there is a dispersal from Indochina to southern Yunnan, the blue area is more than 50% meaning it originated in Indochina. Changed the spaces on both sides of = throughout the text.

L257-258: “A: Australasia; I: Indochina; O: Australasia.” what do you mean?

Those are the areas for the genera outside Eleutharrhena and Pycnarrhena which was not our focus. We added the explanation in the legend to say the areas for the remaining species in Tiliacoreae. 

  1. L260: this is not “DEC” show for the first time.

You mean is different from previous version? Because we changed the sampling the results can be different.

  1. L286: “Figure 4a”, not “Figure 4.a”. “Subpopulation 1”, not “subpopulation 1”

Changed

  1. Figure 4: some font size are obviously too small to read. “Pop x” is used through the text, but “p x” is used in the figure.

Changed pop for p in the text. Size of text increased in figure

  1. L310: no first line indent.

Changed

  1. Figure 5: Capital letters are used in the figure while the figure notes are in lower cases. There is no meaning for a phylogeny tree without supports values, even it is based on morphological data only.

This figure was deleted in the new version

  1. L343: BS and PP in the previous text. And the bootstrap values should be with “%”, change through the text please.

changed

  1. L423: no first line indent.

Added an indent and checked throughout text

  1. Table A1: change the table header as “Taxon, Voucher, Locality, rbcL, atpB, matK, ndhF, trnL-F”. The vouchers and localities should be list as one independent column.

Table changed, voucher and locality added

  1. Figure A1: the PP values should keep two decimal places. And the order of BS and PP in the figure and figure captions should be in the same order.

changed

  1. Figure A2 is not referred in the text. And the p value is bigger than 0.001.

Reference added, yes p is more than 0.001 and the r is less than 0.8. So, the regression correlation is not strong. Therefore, we changed the expression to say there was no correlation.

Try to find a native English speaker to improve your language if possible.

Sorry for the language problems, we submitted this article to a paid service for language editing by a native speaker from MDPI.

Round 3

Author Response

The english language was edited and checked by a native English speaker friend Jo Osborne. 

Fig. 4: “K. Curarea” change to “K. Curarea sp.”

Changed

“D and J were not sampled and added to the analysis due to lack of available DNA sequences.” But I find two available sequences for J. Syrrheonema in the genbank: KX384144 (trnL-F) and KX384080 (matK). It’s very strange. for lack of available DNA sequences.

Sorry we didn’t write clearly, as suggested before by the reviewer we removed Syrrheonema and Pleogyne samples inside the phylogeny because their available sequences were too short and there was no support for them in the phylogeny. We changed this sentence to:

D and J were not sampled and added to the analysis because only few DNA regions were available, and their branches had no support.

And the name of Eleutharrhena macrocarpa CPG29442 is wrong spelled.

Corrected

In the revised manuscript, the authors added the “Table A1. Morphological characters.”

I think there are many problems for the new Table A1 that need to be corrected.

First, the name of Table A1 should change to “Morphological characters and character states used in this study”.

added

Second, for all the content of the table, the first letter should be capitalized (refer to the other tables in the manuscript).

changed

Third, for the first column, “Character numbers” should change to “No.”.

changed

Forth, “male sepals (number)” change to “number of male sepals”. And the same below.

changed